# PTH and the Regulation of Mesenchymal Cells within the Bone Marrow Niche

**DOI:** 10.3390/cells13050406

**Published:** 2024-02-26

**Authors:** Hanghang Liu, Linyi Liu, Clifford J. Rosen

**Affiliations:** 1State Key Laboratory of Oral Diseases & National Center for Stomatology & National Clinical Research Center for Oral Diseases, West China Hospital of Stomatology, Sichuan University, Chengdu 610041, China; hanghangliu_hx@foxmail.com; 2Maine Medical Center, MaineHealth Institute for Research, 81 Research Drive, Scarborough, ME 04074, USA; linyi.liu@mainehealth.org

**Keywords:** PTH, skeletal stem cells, bone remodeling

## Abstract

Parathyroid hormone (PTH) plays a pivotal role in maintaining calcium homeostasis, largely by modulating bone remodeling processes. Its effects on bone are notably dependent on the duration and frequency of exposure. Specifically, PTH can initiate both bone formation and resorption, with the outcome being influenced by the manner of PTH administration: continuous or intermittent. In continuous administration, PTH tends to promote bone resorption, possibly by regulating certain genes within bone cells. Conversely, intermittent exposure generally favors bone formation, possibly through transient gene activation. PTH’s role extends to various aspects of bone cell activity. It directly influences skeletal stem cells, osteoblastic lineage cells, osteocytes, and T cells, playing a critical role in bone generation. Simultaneously, it indirectly affects osteoclast precursor cells and osteoclasts, and has a direct impact on T cells, contributing to its role in bone resorption. Despite these insights, the intricate mechanisms through which PTH acts within the bone marrow niche are not entirely understood. This article reviews the dual roles of PTH—catabolic and anabolic—on bone cells, highlighting the cellular and molecular pathways involved in these processes. The complex interplay of these factors in bone remodeling underscores the need for further investigation to fully comprehend PTH’s multifaceted influence on bone health.

## 1. Introduction

The bone marrow (BM) microenvironment is a highly intricate and dynamic system and consists of a diverse array of cell types and an extracellular matrix (ECM), which provides key signaling events that are crucial for the regulation and maintenance of bone remodeling, development, and hematopoiesis throughout a person’s life [1]. Disruptions in this bone marrow microenvironment can lead to various bone and hematological disorders like osteoporosis and leukemia. It is crucial to understand the orchestrated action of the BM microenvironment for advancing our knowledge of bone cell biology and developing therapies for those diseases.

Parathyroid hormone (PTH), also called parathormone or parathyrin, is an 84-amino-acid polypeptide hormone that is produced and secreted by the parathyroid gland [2]. PTH (1-34), commonly known as teriparatide, holds the distinction of being the first anabolic agent approved by the U.S. Food and Drug Administration (FDA) for the treatment of severe postmenopausal and glucocorticoid induced osteoporosis, due to its central role in the regulation of calcium and phosphorus metabolism [3]. It was observed in the early twentieth century that patients with hyperparathyroidism due to chronically high PTH levels suffered from severe bone disease resulting in multiple fractures. But subsequently, as early as 1923 it was unexpectedly discovered that intermittent PTH (iPTH) could actually stimulate osteogenesis [4]. Now it has been widely accepted that intermittent administration of PTH (i.e., once daily) can promote bone formation within an “anabolic window” and lead to a reduction in osteoporotic fractures [5].

Recently, studies of bone cell metabolic programming have emerged with increasing evidence that a principal impact of PTH is through regulating cellular metabolism. The results of these experiments have shed light on how these actions affect the maintenance of skeletal homeostasis within the bone marrow niche. The current review will cover the detailed mechanisms of PTH on the BM microenvironment including its impact on cellular metabolism.

## 2. Role and Function of Mesenchymal Cells in the BM Niche

The cellular components of the BM niche are highly heterogeneous, such that various genetic approaches, including Cre-mediated lineage tracing and single-cell RNA-sequencing, have made substantial contributions to our comprehension of the cellular architecture and function within the BM niche. In general, only approximately 2% of all BM cells are stromal cellular components, which mainly consist of skeletal stem cells (SSCs), endothelial cells (ECs), osteoblasts, adipocytes, osteocytes and pericytes [6].

### 2.1. Skeletal Stem Cells

SSCs represent a versatile group of multipotent stromal cells capable of differentiating into various cell types. Upon injury or tissue damage, SSCs are able to migrate to sites of inflammation or injury and actively participate in the processes of tissue repair. Single SSC clones have shown the ability to differentiate into multiple mesenchymal lineages, such as osteoblasts, adipocytes, and chondrocytes, when cultured in vitro. Several markers have been used for SSC subgroup identification, including CD146 [7], C-X-C motif chemokine 12 (CXCL12) [8], leptin receptor (LepR) [9], Nestin [10], and neural–glial antigen 2 (NG2) [11]. In addition to multipotency, SSCs can also support hematopoiesis by producing various factors and cytokines that aid in the maintenance and proliferation of hematopoietic stem cells (HSCs) [12]. To be more specific, CXCRL12+ SSCs were found surrounding vascular endothelial cells and express both adipocytic and osteogenic genes; these cells may play a crucial role in maintaining HSC proliferation and survival [13]. Similarly, LepR+ SSCs also contribute to CXCL12 production and partially overlap with CXCRL12+ SSCs. Fate mapping results have shown that LepR+ SSCs can give rise to both osteoblasts and adipocytes, and LepR(+) cells are identified as the primary source for the generation of both osteogenic cells and adipocytes. Besides, LepR+ cells are also required to maintain HSCs’ self-renewal [14,15]. Nestin, an intermediate filament, has been considered to be another characteristic marker of BM SSCs; the entire mesenchymal activity and clonogenicity of pre-reported CD45− SSCs reside within the Nestin+ subset. Nestin+ SSCs are usually present in the periosteum and localize closely to blood vessels. They also show properties of skeletal progenitors in vitro and can form bone in vivo. However, Nestin+ SSCs are neither slow-dividing nor self-renewing in vivo. NG+ positive cells have been referred to as the periarteriolar cell population strongly expressing Nestin. HSCs have been observed to be closely adjacent to the Nestin+ cells, and Nestin+ cells may play an essential role in the control of HSC quiescence and hematopoiesis [6].

### 2.2. Osteoblasts and Osteocytes

Besides SSCs, osteoblasts, adipocytes, ECs, and other mesenchymal lineages are also involved in tissue homeostasis, hematopoiesis support, immune modulation, and tissue repair and regeneration. Osteoblasts and osteocytes are two major bone cells related to bone formation, and the communication between osteoblasts and osteoclasts or HSCs is crucial for bone remodeling and hematopoiesis. Osteoblasts and osteocytes not only produce collagen and minerals, but they also generate numerous molecules like Semaphorin 3A (SEMA3A), Osteoprotegerin (OPG), and the receptor activator of nuclear factor kappa-Β ligand (RANKL). Secreted SEMA3A and RANKL act on osteoclasts (OCs) and regulate the proliferation and differentiation of these cells [16]. Osteocytes may exert a function in the regulation of both SSCs’ and HSCs’ lineage commitments via the mediation of cell senescence [17]. Subsequent investigations have revealed that osteoblasts also play a role in paracrine signaling, mediated by the secretion of various molecules such as Osteopontin (OPN), granulocyte-colony stimulating factor (G-CSF), or Wnt ligands, which in turn regulate HSC activities [18].

### 2.3. Adipocytes

Adipocytes within the BM have been recognized for a long time, but they have been frequently perceived as inert fillers when bone and hematopoietic cells were absent. Recently, mounting evidence suggests that adipocytes within the BM fulfill a dual role. Not only do they act as energy storage units, but they also function as paracrine and endocrine cells. They are capable of influencing and modifying bone metabolism through the secretion of various factors. Translational studies have delved into the connection between BM adipocytes and clinical conditions like osteoporosis, anorexia, and obesity [19,20]. It is now clear that the equilibrium between osteoblasts and adipocytes plays a crucial role in determining an individual’s peak bone mass [21]. Adipocytes can impact SSC differentiation and control the activity of bone-forming cells during bone remodeling, utilizing paracrine and endocrine pathways, particularly through the release of leptin and adiponectin [22]. Moreover, BM adipocytes may release diffusible inhibitors of hematopoiesis and leukocytogenesis [23], and lipids released from adipocytes can also enhance the secretion of pro-inflammatory cytokines by T cells, subsequently triggering a pro-inflammatory cascade, resulting in a negative impact on both the quantity of bone and the overall balance of bone homeostasis [24].

Osteoblasts and adipocytes share a common mesenchymal precursor, as mentioned above. During the initial stages of pre-osteoblast differentiation, various transcription factors like Runt-related transcription factor 2 (RUNX2) and Osterix (OSX) are upregulated. On the other hand, the differentiation of preadipocytes necessitates the involvement of different factors such as platelet-derived growth factor receptor beta (PDGFRβ), CCAAT enhancer binding protein alpha (C/EBPα), Zinc finger protein 423, 467, and 521 (ZFP423, 467, and 521), as well as peroxisome proliferator-activated receptor gamma (PPARγ). In addition, the direction of SSC differentiation toward osteoblasts or adipocytes can be strongly regulated by endocrine molecules, particularly estrogen, Irisin, Leptin, and PTH.

### 2.4. Endothelial Cells (ECs)

Monolayers of ECs form a lining within the lumens of blood vessels in the BM, including arterioles, sinusoids, and an extensive network of capillaries and lymphatic vessels. The network of blood and lymphatic vessels, a vital but sometimes overlooked component of the BM microenvironment, plays a multifaceted role. It not only delivers essential signals for the maintenance and proliferation of hematopoietic stem and progenitor cells within the bone but also regulates the differentiation of perivascular mesenchymal stem cells, guiding their development into bone cells [25]. Recent advancements in imaging techniques, capable of visualizing thick sections of murine bones, have provided significant insights into the diversity and variability of blood vessels within bone tissue [26]. In the past decade, a novel capillary subtype known as “H-type endothelium” has been discovered within the murine skeletal system. These H-type ECs exhibit unique morphological, molecular, and functional characteristics. Specifically, the CD31^hi^/Emcn^hi^ H-type ECs play a significant role in facilitating local vascular growth, generate distinct metabolic and molecular microenvironments, and provide niche signals that are essential for perivascular osteoprogenitors within the BM niche [26].

Through a combination of high-resolution light-sheet imaging and mouse genetics focused on specific cell types, investigators have also confirmed the existence of lymphatic vessels within skeletons. During the process of lymphangiogenesis, the secretion of CXCL12 by actively dividing lymphatic endothelial cells is of utmost importance for both hematopoietic (blood cell-forming) and bone regeneration. More precisely, CXCL12 released by lymphatic endothelial cells plays a role in stimulating the expansion of mature Myh11+ C-X-C chemokine receptor type 4 (CXCR4)+ pericytes, but not Pdgfrb+ cells, Gli1+ cells, and AdipoQ+ mesenchymal cells, which then undergo differentiation into perivascular osteoblasts and contribute to the regeneration of both bone tissue and the hematopoietic system [25].

Summary: Various types of cells in the bone marrow and skeleton regulate bone remodeling.

## 3. The Direct Interactions of PTH with BM Cells

PTH is an essential and key modulator of calcium-phosphate homeostasis, primarily by promoting bone resorption when calcium is scarce and inducing 1,25 dihydroxyvitamin D to promote calcium absorption in the gut. Conversely, intermittent bursts of PTH can stimulate bone formation by encouraging the differentiation of SSCs into osteoblasts. PTH also recruits osteoblast progenitors from bone lining cells and perivascular cells, contributing to bone formation processes [7].

The anabolic actions of iPTH application on bone involve multiple proposed mechanisms, which involve diverse signaling pathways and a range of targets. In general, PTH exerts its action by binding to its specific receptor, PTH1R. Upon binding, PTH activates multiple intracellular signaling pathways, such as the cyclic AMP (cAMP) pathway, as well as pathways involving calcium ions (Ca2+) and protein kinase C (PKC) [27]; these can in turn regulate a multitude of cellular processes. Moreover, there is substantial evidence indicating that the PTH1R is not limited to SSCs and osteoblasts but is also expressed in other cell types, including adipocytes, osteocytes, and ECs, which allows PTH to exert its effects on various cell types. The presence of the PTH1R in multiple cell types within the BM niche suggests a complex and interconnected regulatory network of PTH signaling, contributing significantly to the maintenance of bone homeostasis and supporting the hematopoietic processes.

### 3.1. Skeletal Stem Cells (SSCs) as Targets of PTH

Indeed, several studies conducted over the past two decades have indicated that PTH1-34 plays a role beyond traditional bone remodeling and can also influence osteogenesis and chondrogenesis during bone remodeling and repairing processes. One of the notable effects of PTH1-34 on bone repair is its capacity to stimulate both the proliferation and differentiation of SSCs during the initial stages of bone healing. This effect has been documented in research dating back to 1994 and highlights PTH1-34’s potential as a therapeutic agent for promoting bone regeneration and repair [28].

#### 3.1.1. Proliferation and Osteogenesis of SSCs Induced by PTH

A multitude of studies have indicated that PTH can modestly stimulate the proliferation of SSCs. In addition to one particular study highlighting that PTH impedes the differentiation of cells expressing RUNX2, especially in the context of matrix mineralization [29], most studies have revealed that PTH has a pronounced effect on osteogenic differentiation by enhancing alkaline phosphatase activity and calcium production in a dose-dependent fashion [30,31]. In response to iPTH treatment, LepR(+) MSPCs can differentiate into mature osteoblasts expressing Col1, and the proliferative LepR-Cre-marked lineage cells near bone tissue also increase. The expression level of osteoblast markers including Sp7 and Col1 rises, while adipocyte markers *Cebpb*, *Pparg*, and *Zfp467* decline in the LepR(+) SSC population, indicating a shift towards osteogenic differentiation and reduced adipogenesis due to iPTH treatment [32]. Studies have consistently demonstrated that PTH can have a rejuvenating effect on SSCs isolated from patients with atypical fractures, and these cells exhibit an increase in their capacity to differentiate into bone-forming cells with PTH treatment. PTH appears to enhance both the number and functional activity of osteoprogenitor SSCs [33]. In agreement with a previous study where PTH was shown to increase markers of osteogenic differentiation and promote the migration of SSCs by upregulating the expression of SDF-1 [34], iPTH treatment has also been observed to increase the density of circulating SSCs, characterized by markers CD73+, CD90+, CD105+, CD34−, and CD45−, in women with postmenopausal osteoporosis. Not surprisingly, iPTH treatment has been shown to enhance the in vitro osteogenic differentiation potential of these cells [35].

#### 3.1.2. Adipogenesis and Its Regulation by PTH

The equilibrium between osteoblastogenesis and marrow adipogenesis acts as a crucial regulator for maintaining bone health. This balance serves as a key factor in the prevention or treatment of osteopenia, a condition characterized by lower-than-normal bone density. In addition to its impact on osteogenic lineage commitment, PTH has also been reported to influence the adipogenic differentiation of SSCs. It was first reported by Richard et al. in 2006 that the regulation of adipocyte formation in human SSC cultures by hPTH(1–34) varies depending on its mode of administration. When administered intermittently as a daily 1 h treatment, PTH inhibits adipogenesis primarily through a cAMP-dependent mechanism. However, it is noteworthy that while iPTH does inhibit adipogenesis in SSC cultures, any reciprocal increase in osteoblastic differentiation is less pronounced. Conversely, continuous exposure to PTH does not significantly impact adipogenesis [36]. It has been further confirmed that the osteogenic potential of aged SSCs is enhanced with PTH treatment, while their adipogenesis capacity is attenuated [37]. Parathyroid hormone-related protein (PTHrP) (1-36), another local factor that shares structural similarities with PTH, has also been reported to inhibit adipogenesis in SSCs and increase the gene expression of early-stage osteogenic markers [38]. Therefore, it has also been reported that PTH can fully reverse the suppressed bone formation caused by focal radiation which reduces cell numbers and activity and increases the adipogenic differentiation of SSCs [39]. More recently, we have revealed that marrow adipocytes demonstrate a unique combination of osteogenic and adipogenic characteristics. We also exhibited a distinct responsiveness to PTH and secreted the protein RANKL [40]. Deletion of the PTH1R through Prx1Cre had a profound impact on cell destiny, leading to a significant decrease in osteoprogenitor cells and a substantial increase in marrow adipocytes. Consistently, PTH can directly influence mature adipocytes, triggering lipolysis, which is the breakdown of stored triglyceride and fatty acid. Additionally, these mature adipocytes can respond to PTHrP by altering their adipogenic program, shifting towards a more thermogenic pattern. However, we did not observe any alterations in peripheral fat depots, including subcutaneous or visceral fat with PTH (1-34) treatment, in our study.

#### 3.1.3. Chondrogenesis and Its Regulation by PTH

Alongside its role in steering SSCs towards osteogenic and adipogenic lineage commitment, PTH at lower to moderate concentrations also encourages chondrogenic differentiation, supporting cartilage repair and regeneration. This chondrogenic potential is indicated by the upregulation of critical chondrogenic markers, including SOX9, collagen type II alpha 1 chain (COL2A1), and collagen type V (COL5) [41,42]. However, the knockdown of the transcription factor RUNX1 has been shown to inhibit PTH-induced chondrogenesis, suggesting a pivotal role of RUNX1 in this process [43]. In addition, the effect of PTH on promoting RUNX1 expression and subsequent chondrogenesis is greatly diminished when the PKA signaling pathway is inhibited.

Furthermore, in a Temporomandibular Joint Osteoarthritis (TMJ-OA) model, extracellular vesicles (Exo) derived from SSCs treated with iPTH demonstrate enhanced anti-inflammatory effects, and chondrocytes that co-cultured with iPTH-treated SSCs exhibit improved proliferation, migration, and increased production of ECM components, such as type II collagen (Col-II), compared to the control group [44,45]. Additionally, early intervention with iPTH in osteoarthritis cases has been found to decelerate the degeneration of articular cartilage [46], and alleviate osteoarthritis-related pain by inhibiting sensory nerve growth and Prostaglandin E2 (PGE2) expression in the subchondral bone [47]. Mechanically, iPTH administration significantly reduces the number of matrix metalloproteinase (MMP13)- and phosphorylated suppressor of mothers against decapentaplegic (SMAD)2/3-positive chondrocytes, suggesting that iPTH effectively mitigates the deterioration of condylar cartilage by impeding the transforming growth factor-beta (TGFβ) signaling pathway [48].

#### 3.1.4. PTH and Its Regulation of Angiogenesis

Administration of PTH has led to a notable rise in the number of alpha-Spinal muscular atrophy (αSMA)+ and c-kit+ perivascular/stromal cells, which are likely committed to the osteoblastic lineage. Upon proliferation, these αSMA-positive/c-kit-reactive interstitial stromal cells may differentiate either into a less differentiated cell type that is possibly committed to the osteogenic lineage, or they might develop into vascular smooth muscle cells [49]. Similarly, PTH (1-34) has been demonstrated to directly promote angiogenesis by enhancing the migration of endothelial cells and fostering the formation of blood vessels in vitro [37]. PTH (1-34) also stimulates greater involvement of osteoclasts in bone remodeling by releasing angiogenic and osteogenic growth factors, including CD31, vascular endothelial growth factor receptor (VEGFR), VEGFR2, and von Willebrand factor (vWF), not only in the osteoblasts but also extending to blood vessels and adjacent cells [49,50]. These factors induce early vascularization and indirectly stimulate the migration or differentiation of SSCs, thereby promoting both angiogenesis and osteogenesis. This dual action contributes to the facilitation of fracture healing [51].

#### 3.1.5. Indirect Effect of PTH on Hematopoiesis via Regulating SSCs

PTH also exerts an indirect effect on hematopoiesis through its actions on SSCs. PTH may boost hematopoiesis by activating the IGF system, such as increasing the secretion of insulin-like growth factor (IGF)-2, insulin-like growth factor binding protein (IGFBP)-1, 2, 3, and GM-CSF in SSCs [52]. PTH stimulates the expansion of CD34(+) HSCs and supports hematopoiesis by upregulating the expression level of CDH11 in BM-derived SSCs [53]. Therefore, PTH treatment may serve as an effective strategy to enhance the ability of BM-derived SSCs to support hematopoiesis. Conversely, PTH can also impact CD8+ T cells by increasing their expression of key Wnt factors, including Wnt5a, Wnt10b, Wnt6, and Wnt10a [54]. PTH has been reported to increase the levels of the soluble interleukin-6 receptor (sIL-6R) in primary osteoblast cultures. While sIL-6R does not directly influence osteoblast proliferation or the differentiation of SSCs in vitro, it plays a significant role in the BM microenvironment by enhancing the expansion of myeloid cells and recruiting SSCs via increasing the production of TGF-β1in SSCs [55].

During the development of both mice and humans, MSCs and hematopoietic cells coexist in most hematopoietic tissues, from the embryonic stage to adulthood [56]. This coexistence suggests a close relationship between these two cell types. Numerous studies have shown that the state of HSCs is intimately linked to bone health [57]. Specifically, bone marrow erythroid progenitor cells can influence the differentiation of MSCs into osteoblasts and adipocytes through direct cell-to-cell contact [58]. Rodrigo et al. found that reduced levels of hemoglobin and blood cells might be early indicators of bone fragility in men [59]. In laboratory experiments, it was observed that a low concentration of HSCs in co-culture with MSCs can accelerate the osteogenic progression of MSCs. This acceleration was evidenced by an earlier peak in alkaline phosphatase activity and increased calcium deposition, compared to cultures of MSCs alone [60]. Further studies have confirmed that HSCs can regulate the induction of mesenchymal stromal cells into osteoblasts. This regulation is part of the formation of the stem cell niche and is facilitated by the secretion of Bone Morphogenetic Proteins 2 and 6 (BMP2 and BMP6) [61]. These findings highlight the complex interplay between different cell types in bone health and development. Overall, the indirect effect on hematopoiesis has a beneficial effect on SSCs’ osteogenic differentiation, thereby supporting the differentiation of osteoprogenitors and contributing to the anabolic bone remodeling effects of PTH [54,62,63].

In summary, these findings highlight the pivotal role of PTH1R signaling in directing SSCs towards an osteoblast lineage, steering them away from adipogenic pathways. The underlying mechanisms of this activity and its therapeutic potential are also areas of significant interest.

#### 3.1.6. The Downstream Regulatory Networks of PTH in SSCs

Recently, several downstream signaling pathways of PTH-PTH1R have been reported, including the PKA/PKC pathway, Wnt pathway, TGF-β pathway, BMP pathway, and MAPK pathway (Figure 1).

a.cAMP/PKA pathway and Ca2+/PKC pathway

The PTH1R can engage with two different G proteins to transmit signals inside the cell: the Gs protein and the Gq protein. The Gs protein activates adenylate cyclase and protein kinase A, whereas the Gq protein stimulates phospholipase C, leading to calcium release dependent on inositol trisphosphate (IP3). In SSCs, the primary signaling mechanism behind the anabolic action of PTH involves a gradual increase in intracellular cAMP and its downstream targets, specifically the PKA and cAMP response element-binding protein (p-CREB) [64]. This regulation enhances the proliferation, osteogenic differentiation, and mineralization of SSCs, while concurrently decreasing their adipogenic differentiation [65,66]. In contrast, when treating with a PKA inhibitor, H-89, or knocking-down CREB, the stimulatory effects of PTH on proliferation and osteoblast differentiation in SSCs is blocked [66,67]. Additionally, it has been reported that endogenous prostaglandins can inhibit the pro-osteogenic effects of PTH. It is suggested that endogenous prostaglandin E2 might desensitize the cellular cAMP responses to PTH, thereby counteracting the anabolic actions of PTH [68]. This highlights the critical role of the cAMP/PKA signaling pathway and p-CREB in conducting the beneficial action of PTH on SSCs.

Conversely, PTH’s catabolic effect on SSCs is mediated by Ca2+ oscillations induced by PKC [69]. It is probable that PTH1R in SSCs engages in dynamic interactions with both Ca2+-dependent and cAMP-dependent signaling pathways, and the dominance of one pathway over the other determines the specific functional impact of PTH on these cells. In essence, the choice between these signaling pathways determines whether PTH promotes osteogenesis or exerts an anti-osteogenic effect in SSCs. However, it has also been reported that iPTH leads to an increase in PKC activity at day 7 during osteogenic differentiation, resulting in a remarkable increase of osteogenic genes and proteins in SSCs, whereas the inhibition of PKC activity attenuates these effects [70]. Thus, it is still not clear what the role of the Ca2+/PKC pathway is in the PTH-regulated osteogenic/adipogenic differentiation of SSCs in the BM niche.

b.Wnt signaling pathway

Wnt signaling in skeletal tissue operates through the binding of Wnt ligands to receptors such as the fizzled or/and low-density lipoprotein receptor family of protein 5/6 (LRP5/LRP6), as well as frizzled receptors primarily found on the cell surface. When a ligand binds to these receptors, it leads to the stabilization and nuclear translocation of β-catenin. This translocated β-catenin then activates the transcription of Wnt target genes that play a crucial role in supporting bone formation.

Current studies reveal that both PTH and PTHrP can induce the expression of IGF-I and subsequently activate the downstream Wnt pathway, thereby leading to an increase in bone mineral density and a reduction in the incidence of both vertebral and non-vertebral fractures [71]. Likewise, Yukata et al.’s study yielded similar results, demonstrating that PTH increases the expression of Cyclin D1 in periosteum SSCs. The periosteum is a primary source of stem cell populations involved in bone regeneration. Additionally, PTH leads to increased staining of active β-catenin in osteoblast populations derived from SSCs, resulting in the enhanced proliferation and expansion of the periosteal stem cell population, ultimately accelerating bone formation and the healing of fractures [72].

c.TGF-β signaling pathway

iPTH has been found to upregulate the transcription of genes related to osteogenesis and the expression of IGFBP7 in both SSCs and pre-osteoblasts. IGFBP7 is believed to inhibit the mammalian target of rapamycin (mTOR) pathway while enhancing the TGF-β/bone morphogenetic protein (BMP) and canonical Wnt signaling cascades [73]. However, in an osteoarthritis model with excessive active TGF-β signaling, iPTH plays a crucial role in maintaining coordinated subchondral bone remodeling as a negative feedback break. iPTH mitigates the increases in both Nestin-positive and Osterix-positive cells within the BM and induces the formation of osteoids on the surface of subchondral bone. When iPTH is initiated early, it helps attenuate abnormal bone remodeling in osteoarthritis patients by regulating the recruitment and migration of SSCs in the BM. Mechanistically, on the one hand, PTH was found to inhibit the phosphorylation level of SMAD3, a protein that can bind to the p16(ink4a) gene promoter region. This inhibition leads to a reduction in the accumulation of senescent cells and an increase in the cellular proliferation of SSCs [74]. On the other hand, in response to PTH, PTH1R and the transforming growth factor-beta receptor II (TbRII) form an endocytic complex. PTH induces the endocytosis of TbRII, leading to a decrease in TGF-β1-induced phosphorylation and the nuclear translocation of its downstream targets, SMAD2 and SMAD3, when compared to the control group. This suggests a key role of PTH1R signaling in sustaining subchondral bone repair by inhibiting excessive active TGF-β signaling [47].

d.BMP signaling pathway

The administration of iPTH in OVX mice, as first described by Ogtia et al. in 2008, leads to the rapid phosphorylation of SMAD1/5/8, which are downstream targets of BMP-2, in the periosteum. This, in turn, increases the osteogenic differentiation of SSCs [75]. iPTH has also consistently been found to activate the SMAD1 pathway indirectly via inducing the endocytosis of LRP6. LRP6, a co-receptor in the Wnt signaling pathway, has been identified to interact with both the PTH and BMP extracellular signaling pathways. It forms a complex with the PTH1R and shares common antagonists with BMP signaling, suggesting cross-talk or a shared regulation between these pathways. Treatment with PTH induces the endocytosis of the PTH1R-LRP6 complex and leads to an enhancement in the phosphorylation of SMAD1, resulting in the enhanced differentiation of Sca-1(+) CD45(−) CD11b(−) SSCs into the osteoblast lineage. Importantly, this effect is abolished when PTH1R, or β-arrestin, is deleted or inhibited, indicating the involvement of these components in mediating the phosphorylation of SMAD1 in response to PTH treatment [76].

e.MAPK pathway

Additionally, PTH has been found to reverse inadequate osteogenesis and excessive oxidative stress in SSCs treated with high glucose and palmitic acid or lipopolysaccharide (LPS). Mechanistically, PTH activates both the p38 and c-Jun N-terminal kinases-mitogen-activated protein kinases (JNK-MAPK) signaling pathways by upregulating the phosphorylation of their respective subunits. It has been observed that a selective inhibitor targeting either p38 or JNK-MAPK signaling can weaken the effects of PTH on promoting increased osteogenic differentiation in SSCs [77,78]. However, in osteoarthritis (OA) patients, PTH (1-34) has been observed to inhibit the expression of type X collagen (COL10) and stimulate the expression of COL2 in SSCs. These effects are sustained over time and are related to the decrease in the phosphorylation of the p38 and AKT protein kinase signaling pathways [79].

iPTH has also been observed to indirectly influence the behavior of SSCs by increasing the release of Amphiregulin from osteoblastic cells and osteocytes. Amphiregulin is a growth factor that, once secreted, can engage with epidermal growth factor receptors (EGFRs) present on SSCs, stimulating the Akt and p38-MAPK pathways within the SSCs and promoting cell survival, proliferation, and migration [80]. However, controversial results were obtained from Jay’s study where iPTH treatment of Amphiregulin-knockout (KO) mice also led to increased bone formation, and the bone anabolic effects of PTH may not require Amphiregulin [81].

f.Key effectors of PTH action

Interestingly, recent studies have identified ZFP467 as a newly discovered key downstream effector of PTH in SSCs. ZFP467 was found to be quickly downregulated by PTH. ZFP467 has the ability to activate a PPAR response element reporter construct and recruit a histone deacetylase complex [82]. Notably, ZFP467 serves as a novel co-factor that promotes the differentiation of SSCs into adipocytes, as indicated by the upregulation of adipocyte-related markers like C/EBPα, adiponectin, and resistin. Conversely, ZFP467 suppresses the differentiation of SSCs into osteoblasts, as evidenced by the downregulation of markers associated with late osteoblasts and osteocytes, such as osteocalcin and sclerostin [82].

In line with previous findings, our recent study reaffirmed that ZFP467 has a detrimental impact on skeletal homeostasis and promotes adipogenesis of SSCs. When ZFP467 is globally deleted, it leads to an increase in PTHR1, cAMP levels in SSCs, and promotes bone turnover [83]. Mechanistically, our research has further revealed that PTH1R and ZFP467 form a positive feedback loop that promotes PTH-induced osteogenesis. When ZFP467 is conditionally deleted in Prx1 positive osteogenic precursors, it results in higher bone mass in mice. The loss or PTH1R-induced downregulation of ZFP467 activates a pathway that upregulates the expression of PTH1R via nuclear factor kappa b subunit 1 (NFκB1) in SSCs. This, in turn, enhances cellular responsiveness to PTH and leads to increased bone formation [84].

Nuclear matrix protein 4 (Nmp4) is another key effector that may mediate the downstream targets of PTH signaling in SSCs. Mice with a global knockout of the Nmp4 gene exhibit an enhanced response to PTH resulting in increased trabecular bone, but they do not display any noticeable skeletal abnormalities under baseline conditions. In the case of mice with the Nmp4 gene conditionally deleted in Prx1-expressing cells (Nmp4(fl/fl);Prx1Cre(+)), they closely resemble the global Nmp4 knockout mice in terms of their femur bone structure. However, the conditional loss of Nmp4 from the mature osteoblasts and osteocytes (Nmp4(fl/fl); BglapCre(+) or Dmp1Cre(+)) failed to increase bone volume or enhance PTH response. These findings suggest that Nmp4 in Prx1-expressing SSCs plays a key role in regulating the improved response to PTH therapy. Additionally, Nmp4 exhibits stage-specific effects on osteoanabolism, with its influence varying depending on the specific cell type and stage of bone development [85].

g.Other modulators of PTH

In addition to the anabolic effect of PTH on the differentiation of SSCs, PTH also dramatically enhances the migration and adhesion of these cells by upregulating the protein level of several cytokines or related receptors, including fibronectin, CXCR4, C-C motif chemokine receptor 2 (CCR2), and intercellular adhesion molecule 1 (ICAM-1). Specifically, when SSCs undergo inhibition of the AK mouse plus Transforming (AKT) or mTOR complex 1/2 (mTORC1/2) signaling pathway, or when Rictor is silenced, there is a significant decrease in the cell migration and adhesion induced by PTH. However, this effect is not observed when the SSCs are subjected to the inhibition of the mTORC1 pathway alone or through raptor silencing. Therefore, it is evident that PTH facilitates the migration and adhesion of SSCs via the Rictor/mTORC2-AKT signaling pathway in an in vitro setting [86].

Consistent with most studies where PTH had a limited effect on SSC proliferation, PTH has also been shown to promote the proliferation of SSCs without affecting apoptosis but reducing cell senescence. Reactive oxygen species (ROS) are chronic and persistent damaging agents that play a significant role in the aging process, and in samples treated with PTH, the percentage of cells positive for oxo8dG (a marker associated with oxidative DNA damage) is significantly lower, indicating that PTH treatment helps reduce oxidative DNA damage in SSCs [87].

Summary: Several key pathways and processes in SSCs are modulated by PTH. These include osteogenesis, angiogenesis, adipogenesis, and lymphagniosis. Cell networks activated by PTH include cyclic AMP MAPK and the BMP pathways.

### 3.2. Osteoblasts and Osteocytes as Targets

The levels of PTH in the bloodstream can lead to either catabolic (bone-resorbing) or anabolic (bone-forming) effects on bone, depending on the timing and pattern of the hormone’s elevation. The anabolic effects of iPTH treatment are attributed to several concurrent mechanisms. These include the activation of immediate-early genes, a rise in the expression and/or function of key osteoblast transcription factors, and the suppression of proteins that negatively impact osteogenesis, like sclerostin. In contrast, continuous PTH administration primarily causes bone loss by increasing the expression of RANKL, a promoter of osteoclast activity, and decreasing the expression of osteoprotegerin, which is a natural inhibitor of bone resorption [88].

Moreover, it is well known that PTH and PTHrP are key regulators in calcium and phosphate metabolism, as well as bone remodeling. Similar to SSCs, upon the binding of PTH or PTHrP to the PTH1R receptor, a cascade of specific responses is initiated at the level of G proteins, subsequently triggering different biological responses. While the activation of most PTH-responsive genes is a result of the combined action of Gs, Gq, and G12/13 proteins, a distinct set of genes is regulated by the trio of Gs, Gq, and G12/13, including the Gsα-cAMP-PKA pathway, Gq-PLC-β-PKC pathway, and Gα12/13-phospholipase D-transforming protein RhoA pathway [89,90]. It has been reported that varying effective domains within PTH or PTHrP are responsible for the differential activation of the PTH1R receptor, which in turn selectively triggers downstream signaling pathways [91,92,93,94]. Overall, this interaction of PTH and PTHrP with different PTH1R conformations may help explain the differences in signaling pathways leading to a predominant anabolic and catabolic biological response.

(1)Cellular actions of PTH contribute to increased bone formation: anabolism of bone

Both animal and clinical studies have found that iPTH is effective in promoting bone formation. Daily administration of PTH, as well as PTHrP, injections has been found to reduce apoptosis in osteoblasts and osteocytes while simultaneously increasing the number of osteoblasts [95,96,97,98]. This leads to a higher activation frequency and rate of bone formation, culminating in improved bone mineral density and enhanced bone strength [99].

#### 3.2.1. Osteoblasts

PTH enhances bone mass by increasing the osteoblast population, which is achieved through promoting the generation of new osteoblasts, decreasing the rate of osteoblast apoptosis and promoting DNA repair, or through a combination of these actions [100]. It has been noted that PTH mitigates the negative impact of glucocorticoids on osteoblast survival and the Wnt signaling pathway [101], and treatment with PTH has been shown to inhibit osteoblast apoptosis in bones affected by diabetes [102].

Additionally, PTH has the ability to activate genes responsible for osteogenesis by inducing the expression of specific transcription factors. It can also enhance the multiplication and maturation of osteoblasts via upregulating the full expression of two important osteogenic effectors, OPN and bone sialoprotein (BSP) [103,104]. Furthermore, PTH can initiate the conversion of lining cells into osteoblasts and slow down the progression of osteoblasts back into lining cells [2,105,106]. The processes that promote osteogenesis are multifaceted, including the activation of osteoprogenitor cells or osteoblasts and the initiation of mineralization [107]. This leads to the enhanced activity of osteoblasts, reduced death of these bone-forming cells, heightened PKA/PKC signaling, Wnt/β-catenin signaling, MAPK signaling, and adjustment of the RANKL/OPG pathway, all contributing to the upsurge in bone formation [108] (Figure 2).

a.The downstream regulatory network of PTH action in osteoblasts

PKA and PKC pathwaysPKA pathway

PTH acts on PTH1R to increase cAMP accumulation in the osteoblast in a dose-dependent manner, and the subsequent activation of PKA occurs swiftly upon initiation, hitting its peak activity within 30 to 60 s. This activation is evident even before there is a detectable rise in the overall cAMP levels within the cell [91]. Over the past decades, multiple downstream targets of the cAMP-PKA signaling pathway have been identified in response to PTH treatment.

Following PTH signaling, nascent polypeptide-associated complex and coactivator alpha (αNAC) is identified as a substrate for PKA. αNAC operates as a transcriptional co-factor within osteoblasts, influencing the transcription of target promoters that are regulated by basic leucine zipper (bZIP) transcription factors. The phosphorylation of αNAC by PKA facilitates its movement into the nucleus where it accumulates at specific target gene promoters, playing a role in regulating transcription. This regulatory mechanism is significant in affecting bone mass. Experimentally, it has been demonstrated that PKA can phosphorylate αNAC at serine residue 99 in vitro, results in its interactions with c-Jun transcription factors, enhances activator protein 1 (AP1) transcriptional activity, and leads to the accumulation of αNAC at the promoter region of the target gene osteocalcin (Ocn), which is involved in bone formation [109]. Moreover, it has been observed that JunD, one of the bZIP family transcription factors, and αNAC are both present on the proximal promoter of LRP6, a co-receptor for the PTH1R which is crucial for the optimal activation of PTH signaling. When αNAC is activated by PTH, it enhances the transcriptional activity of JunD on the Lrp6 gene [110]. Nuclear factor interleukin-3-regulated (Nfil3) and Ubiquitin specific peptidase 53 (Usp53) have also been identified as target promoters of cAMP-PKA-αNAC pathways. Upon stimulation by PTH, the phosphorylated CREB-JunB complex attaches to the proximal promoters of Nfil3 (−818/+182 bp) and Usp53 (−2325/+ 238 bp) within osteoblasts in the presence of αNAC, potentially leading to an increase in osteoblastic activity and thereby promoting bone formation [111,112].

In addition to the previously mentioned targets, Fibroblast Growth Factor 23 (FGF23), Sodium-Hydrogen Exchange Regulatory Factor 1 (NHERF1), and Transient Receptor Potential Vanilloid 4 (TRPV4) have also been identified as downstream targets of the cAMP-PKA signaling pathway in response to PTH, which contribute to the anabolic, or bone-building, actions of PTH. Specifically, NHERF1 plays a role in the regulation of PTH effects on sodium-dependent phosphate (Pi) transport. This regulatory action of NHERF1 on PTH varies across different stages of osteoblast development, influencing Pi transport in a manner that is dependent on the specific phase of osteoblast proliferation and maturation [113]. In addition, TRPV4 is implicated in facilitating the entry of extracellular calcium into osteoblasts, which is crucial in the cellular responses to PTH that involve both structural changes and cell movement [114].

FGF23, a key osteocyte-derived regulator of phosphate homeostasis and bone mineralization, shows time- and dose-dependently increased expression levels in osteoblasts. The regulation of FGF23 mRNA expression is specifically mediated through the cAMP-PKA signaling pathway, rather than the IP3-PKC pathway within osteoblasts/osteocytes [115,116,117]. Following the upregulation of FGF23 during iPTH treatment, the gene Calca, which encodes procalcitonin (ProCT), is also found to be upregulated [118]. ProCT is emerging as a novel target gene within the FGF23 signaling network and PTH’s regulatory scope, with a role that extends to the inhibition of osteoclast formation, and thereby limits bone resorption during iPTH treatment.

The PTH1R-cAMP-PKA pathway, central to the cellular actions of PTH, has been found to be modulated by other receptors, including the β2 Adrenergic Receptor (β2-AR). This interaction suggests cross-talk between different receptor systems, extending the complexity of how PTH influences bone metabolism. The research of Hanyu et al. in 2012 indicates that iPTH lacks osteoanabolic activity in mice that do not have the β2-AR. It was discovered that the deficiency of β2-AR prevents the expression of iPTH-targeted genes that are involved in bone formation and resorption and are usually regulated by the cAMP/PKA signaling pathway [119]. Intriguingly, PTH treatment has been shown to quickly suppress the expression levels of β2-AR mRNA in the study of Moriya et al. [120], which subsequently leads to an increase in the phosphorylation of the CREB. Actually, β2-AR has been reported to exert direct negative effects on bone mass by activating bone resorption and suppressing bone formation. One of the potential factors involved in the lack of PTH anabolism seen in β2-AR-KO mice could be due to the downregulation of genes encoding proteins necessary for the osteoblastic function from other cell types [119]. These kind of intriguing properties and links between PTH treatment and the β2-AR may also be one of their ways of desensitization and may serve as a break for PTH signaling. Therefore, the β2-AR may play a critical role in regulating the effects of the PTH1R-cAMP-PKA pathway on bone metabolism, but the exact mechanism is still unclear.

Furthermore, PTH and PGE2 exhibit comparable effects on bone remodeling. It is hypothesized that PTH may induce local bone effects through upregulating cyclooxygenase-2 (COX-2) and nuclear receptor related 1 (NURR1) expression, which leads to increased production of PGE2. PTH activates the cAMP-PKA pathway and facilitates the nuclear translocation of the transcription factor NFATc1, enhancing COX-2 and NURR1 promoter activity [121,122]. However, in a regulatory feedback mechanism, BM macrophages respond to COX-2 and secrete an inhibitory factor when stimulated by RANKL. This factor hinders the PKA pathway’s stimulation of osteoblastic differentiation by PTH [123]. This indicates that while PTH promotes bone formation, it also initiates a negative feedback loop through the induction of RANKL and COX-2 expression, which can counteract its anabolic actions, thus maintaining a balance in bone remodeling processes.

PKC pathway in osteoblasts

PTH regulates important genes involved in bone remodeling primarily through the cAMP-PKA signaling pathway. Conversely, the PKC signaling pathway is not essential, and might even act as an inhibitor, to the osteoanabolic activities of PTH [124]. Despite the general understanding that PKC signaling may not be necessary for the osteoanabolic actions of PTH, there are studies suggesting otherwise. These studies have demonstrated that PTH can indeed stimulate the proliferation of osteoblastic cells, with this effect being reliant on the activation of PKC. The signaling through PKC by the PTH1R has been indicated as significant for the upregulation of Cyclin D1 expression—a protein essential for cell cycle progression—and consequently for the proliferation of osteoblastic cells [125].

Wnt pathway in osteoblasts

Low-dose administration of PTH has been observed to elevate the expression of specific markers associated with mature osteoblasts. Alongside this, there’s an upregulation in the effectors of the Wnt signaling pathway, such as β-catenin [72,126,127,128], WISP1 [129,130], and USP2 [131], which are important for the growth and repair of bone, and subsequently increase Cyclin D1 and RUNX2 expression. Furthermore, in osteoblastic cell cultures, iPTH significantly decreases the expression of DKK1 [132,133] and WASF2 [134], two known inhibitors of Wnt signaling which are essential for bone formation. Consistently, the constant activation of the PTH/PTHrP receptor in differentiated osteoblasts significantly boosts the levels of key Wnt signaling molecules, including LRP5, Wnt7b, and Wnt10b, leading to increased recruitment, proliferation, and differentiation of osteoblasts [135].

PTH also induces the phosphorylation of β-catenin through a PKA-dependent mechanism [129] or a signal transducer and activator of transcription (STAT) 3-dependent [136] mechanism and substantially increases the expression of Lef1, a transcription factor activated by Wnt signaling. The resulting regulation of BSP expression may contribute to the antiapoptotic and pro-survival actions on osteoblasts of PTH.

In vivo experiments have further confirmed that PTH is effective in promoting the repair of DNA double-strand breaks (DSBs) in osteoblasts that have been exposed to radiation. This repair mechanism is facilitated through the activation of the β-catenin pathway, which in turn accelerates the repair of bone that has been damaged by radiation [137]. Moreover, a pre-treatment strategy that activates β-catenin in osteoblasts suggests that the activation of β-catenin may prime osteoblasts to respond more effectively to PTH in a preclinical model of T1DM, thereby optimizing bone repair and strengthening processes [138].

TGF-β pathways in osteoblasts

It was reported by Hisa et al., in 2011 that PTH activates the expression of the transmembrane protein 119 (TMEM119) via SMAD3 signaling within osteoblasts. TMEM119 is found to interact closely with SMAD1, SMAD5, and RUNX2, and promotes the osteogenic differentiation of osteoblasts [139]. Additionally, following treatment with PTH, osteoblasts become the primary cells expressing the chemokine MCP-1. MCP-1 plays a crucial role in recruiting monocytes, which then differentiate into macrophages and osteoclasts. Since the bone matrix is a rich source of latent TGF-β, the short-term creation of osteoclasts elevates the release of active TGF-β from the bone matrix, thereby boosting TGF-β signaling in surrounding osteoblasts and initiating new bone formation [140].

However, similar to SSCs, TβRII interacts with the PTH1R receptor and forms a complex. Upon PTH stimulation, the TβRII–PTH1R complex is internalized, which in turn diminishes the signaling efficiency of both PTH and TGF-β. Thus, it was reported that PTH1R signaling inhibits TGFβ signaling during the process of osteoblastic differentiation [48]. Furthermore, TβRII has the ability to directly phosphorylate the cytoplasmic domain of PTH1R. This phosphorylation event is a regulatory mechanism that modulates the PTH-induced endocytosis of the TβRII–PTH1R complex, thereby influencing the signaling outcomes in osteoblasts [141,142,143]. Deletion of TβRII in osteoblasts leads to an increase in the cell-surface expression of PTH1R and enhances PTH signaling.

IGF1-IGF1R signaling in osteoblasts

IGF1 is a critical downstream effector of the PTH1R-PKA signaling pathway. It plays a significant role in bone biology, as it not only promotes the osteogenic differentiation of SSCs but also enhances the functional activity and mineralization of mature osteoblasts. Osteoblasts, when stimulated by iPTH, secrete IGF1, which then activates insulin receptor substrate-1 (IRS-1) in osteoblast precursors found in the BM, leading to bone formation [144]. Thus, the anabolic effects of PTH are closely associated with both paracrine and autocrine actions of IGF1 [145]. Moreover, PTH and IGF1 together synergistically promote the differentiation of osteoblasts into osteocytes. At the molecular level, a specific tyrosine residue (Y494) that is located on the cytoplasmic domain of PTH1R can be phosphorylated by the IGF1R. This phosphorylated PTH1R is found to associate with the barbed ends of actin filaments, a crucial step in actin polymerization. This process is essential for the morphological transformation of osteoblasts into osteocytes [146].

MAPK pathway in osteoblastsExtracellular Signal-Regulated Kinases (ERK) pathway

In addition to the above mentioned, that PTH1R endocytosis leads to the activation of ERK1/2, PTH can also rapidly, though transiently, increase the phosphorylation of ERK and Akt in osteoblasts through a PKA-dependent mechanism [147,148]. iPTH has been reported to promote osteogenesis through pathways reliant on both PKA and ERK1/2, with the inhibition of the PKA pathway leading to a reduction in iPTH-induced ERK1/2 phosphorylation [149].

Conversely, both PTH and PTHrP have been shown to regulate MAPK phosphatase 1 (MKP1) and subsequently downregulate pERK1/2, p-p38, and Cyclin D1 in the maturation stage specific to osteoblasts [150]. Mechanistically, PTH or PTHrP treatment increases the expression of mitogen-activated protein kinase (MKP)-1 in differentiated osteoblasts [147], and (MKP)-1 has been identified to induce growth arrest in osteoblasts and promote their mineralization by inactivating pERK1/2 and downregulating Cyclin D1 [151], indicating its role as a mediator of PTH1R’s anabolic actions in cell cycle regulation and differentiation of mature osteoblasts [152].

p38 pathway

iPTH stimulates the production of cAMP, which then activates PKA. PKA, in turn, activates various downstream signaling molecules, including the p38-MAPK, specifically in mature osteoblasts. The activation of p38-MAPK by PTH signals promotes the transcriptional activity and function of osteoblasts, thereby enhancing bone formation and contributing to the overall increase in bone density observed with iPTH therapy [153]. Specifically, cell cycle and apoptosis regulatory protein (CARP)-1 is identified as another novel transducer of signaling pathways that regulate cell growth and differentiation. PTH has been observed to utilize an ERK-independent but PKA- and p38-dependent pathway to downregulate (CARP)-1 expression levels in the osteoblastic lineage, and thereby promoting cell survival and increasing mineralization [154]

MMP13 pathway in osteoblasts

MMP13, also known as collagenase-3, is crucial for osteoblast maturation and is involved in various pathological conditions. MMP13’s transcription is influenced by a plethora of growth factors and hormones, such as vitamin D hormone (1,25(OH)2D3), PTH, and various cytokines [155]. PTH has been found to upregulate the expression of MMP13 in osteoblasts, with MMP13 playing an integral role in the process of bone remodeling [156]. The transcriptional regulation of MMP13 involves histone deacetylases (HDAC) 4, which interacts with the transcription factor RUNX2 and can bind to and repress the MMP13 promoter [157]. PTH can induce phosphorylation of HDAC4 via PKA, particularly at serine 740 and serine 355. This phosphorylation leads to the dissociation of HDAC4 from the RUNX2 at the MMP13 promoter, thereby facilitating MMP13 gene transcription [158,159].

Moreover, the p300/CBP-associated factor (PCAF) also plays a role in the transcriptional activation of MMP13 by PTH. Lee et al. reported that PTH treatment enhances the acetylation of PCAF in a p300-dependent manner and recruits PCAF to the proximal promoter of MMP13, where it cooperates with RUNX2 to upregulate MMP13 transcription [160]. Myocyte enhancer factor 2c (MEF2C), another transcription factor, has been found in association with HDAC4 on the MMP13 promoter. Following PTH treatment, this association is reduced, allowing MEF2C to partner with AP-1 and PCAF to stimulate transcription [161].

On the contrary, Sirtuin 1 (SIRT1) acts as a feedback inhibitor of MMP13 transcription. PTH treatment increases the interaction of SIRT1 with c-Jun, leading to the deacetylation of c-Jun. Subsequently, the SIRT1/c-Jun complex is recruited to the AP-1 site on the MMP13 promoter, which results in the inhibition of PTH-stimulated MMP13 expression [162].

Endoplasmic reticulum (ER) stress and oxidative stress

Interestingly, PTH was also found to be involved in the regulation of ER and oxidative stress in recent studies. PTH has been found to potentially enhance the interaction between heat shock protein 90 (HSP90) and PKR-like endoplasmic reticulum kinase (PERK), contributing to the stabilization of the PERK protein during the early phase of PTH-induced ER stress [163]. Activating transcription factor 4 (ATF4) has been recently considered to be one of the downstream targets of PTH-PKA [164] and the PERK-eukaryotic translation initiation factor 2a (eIF2α)-ATF4 [163] signaling pathway in osteoblasts. ATF4, a stress response protein, quickly upregulates the mRNA and protein expression of FGF2 [164,165] and SLPI [164], but not FGF3 [166], resulting in promoted osteoblast differentiation and enhanced adhesion of osteoblasts to neighboring osteoclasts. This increased adhesion may serve as a mechanism by which osteoblasts can locally inhibit osteoclastic bone resorption [163].

Additionally, PTH also decreases hypoxia-inducible factor-1 alpha (HIF-1α) in an HSP90-dependent manner [167]. The bone anabolic effects of PTH are, in part, attributed to its capacity to mitigate oxidative stress in osteoblasts. PTH has been found to decrease the production of reactive oxygen species (ROS) and the activation of forkhead box protein O (FoxO) following oxidation induced by hydrogen peroxide (H₂O₂) [168,169]. This reduction in ROS and FoxO activation is associated with a decrease in lipid peroxidation, caspase-3 activation, and overall oxidative damage within these cells. At the molecular level, PTH reduces ROS levels and the phosphorylation of p66(Shc), downregulates the expression of the lipoxygenase Alox15, and increases glutathione levels, leading to a decrease in FoxO transcriptional activity and the inhibition of osteoblast apoptosis [147,168]. Conversely, the absence of PTH1R signaling in mature osteoblasts/osteocytes correlates with an increase in markers of oxidative stress and cell senescence, such as p16(Ink4a)/Cdkn2a and 4-hydroxy-2-nonenal [170].

Other pathways induced by PTH

L-type calcium channels, typically expressed in osteoblasts but found at lower levels in osteocytes, and T-type calcium channels, more commonly associated with osteocytes, have been identified as mediators of PTH effects in these cells independent of the PKA or PKC pathways [171]. Furthermore, the calcium sensing receptor (CaSR) has been implicated in the anabolic effects of PTH on bone [172]. Additionally, the ClC-3 chloride channel is also involved in promoting osteogenesis through the RUNX2 signaling pathway in response to PTH. The presence of ClC-3 chloride channels in osteoblasts enhances their responsiveness to PTH, leading to osteogenic differentiation [173].

PTH has also been shown to regulate the expression of p62, a multifunctional protein that plays a role in signaling and autophagy. PTH promotes osteoblast differentiation by increasing p62 levels, which then contributes to the stimulation of RUNX2. In contrast, the osteogenic effects of PTH are significantly compromised in osteoblasts that lack p62, indicating that p62 is essential for PTH’s action on these cells [174]. Recent observations have highlighted the co-localization of PTH1R with primary cilia, an antenna-like structure on the cell surface which is involved in cellular signaling processes, in osteocytic and osteoblastic cells upon PTH stimulation. Furthermore, the activation of PTH1R by PTH has been shown to increase the expression of Gli-1, a transcription factor that is part of the Hedgehog signaling pathway, in both osteocytes and osteoblasts, suggesting that PTH1R might exert its pro-survival effects through the Hedgehog pathway [175].

Summary: PTH targets multiple cells in both the bone marrow and in the skeleton to induce an anabolic response and increased bone formation.

b.Cellular processes and other regulatory mechanisms mediated by PTH in osteoblasts

PTH1R endocytosis

Upon ligand engagement, PTH1R is phosphorylated by G protein-coupled receptor kinases (GRKs) at its C-terminus or the third intracellular loop [90]. This phosphorylation triggers the recruitment of β-arrestin, leading to G protein receptor uncoupling and subsequent internalization of the PTH1R through endocytosis [176].

The internalized β-arrestin–PTH1R complex can either quickly re-emerge to the cell surface if transient, or slowly recycle back or be directed to lysosomal degradation if stable, affecting the number of membrane receptors [177]. Within the cell, β-arrestin serves as a scaffold for endocytosis-related proteins and diverse signaling molecules, thus connecting PTH1R to a variety of pathways, including prolonged cAMP signaling [178] or activation of the p38-MAPK and ERK 1/2 pathways [179].

Recently, it has been observed that a β-arrestin signal biased agonist, derived from modified PTH, has been found to promote the proliferation and survival of osteoblasts. In addition, the interaction between β-arrestin and Cx43 has been identified as significant for the modulation of cAMP signaling pathways. When Cx43 binds to β-arrestin, it sequesters β-arrestin away from GPCRs and facilitates enhanced cAMP signaling [180], thereby exerting a permissive role on osteoblast survival induced by PTH [181].

Bioenergetic pathways activated by PTH

PTH triggers significant metabolic shifts in osteoblasts, notably enhancing the uptake and oxidation of glutamine and fatty acids, processes believed to be essential for the hormone’s role in osteoblast function [182,183]. The uptake of [1-(14)C]-2-deoxy-D-glucose (2DG) and glycogen synthesis in osteoblasts is also elevated following PTH exposure [184]. Furthermore, the deletion of glutaminase 1 (GLS1), an enzyme critical in glutamine metabolism, has been shown to completely inhibit the osteoanabolic response induced by PTH [182]. Conversely, osteoblasts are known to utilize aerobic glycolysis during differentiation. PTH has been observed to stimulate glucose consumption and lactate production even in the presence of oxygen in osteoblasts, while concurrently inhibiting the flow of glucose into the tricarboxylic acid cycle. At the molecular level, the enhanced aerobic glycolysis has been found to be a downstream effect of IGF1 signaling induced by PTH and is contingent upon the activation of the mammalian target of mTORC2 [185]. Lactate, the end product of glycolysis, further induces the formation of ALP-positive osteoblasts, heightens ALP activity, and upregulates the expression of osteogenic markers via augmenting the activation of CREB to its target genes in PTH-treated osteoblasts through the GPR81-Gβγ-PLC-PKC-Akt signaling pathway [186].

Micro-RNA

PTH has been consistently shown to enhance differentiation and glucose uptake in osteoblasts. The increase in glucose uptake is linked to a dose-dependent stimulation of miR-451a expression in these cells. PTH also suppressed AMP-activated protein kinase (AMPK) phosphorylation through the phosphoinositide 3-kinases (PI3K)-mTOR-AKT signaling axis, which in turn prevents the phosphorylation and inactivation of octamer-binding transcription factor 1, which is known to act on the promoter region of miR-451a. Consequently, PTH facilitates the upregulation of miR-451a, which drives osteoblast differentiation by suppressing Odd-skipped related 1 (Osr1) and activating the transcription of RUNX2 [187]. Additionally, PTH has been reported to increase the expression of the long non-coding RNA (lncRNA) SUPT3H-1:16. The lncRNA SUPT3H-1:16 acts as a molecular sponge for miR-6797-5p, protecting RUNX2 from the inhibition of miR-6797-5p. This protective sponging mechanism allows RUNX2 levels to be maintained, thereby promoting osteoblast differentiation [188].

#### 3.2.2. Osteocytes and PTH

Osteocytes, which are transformed osteoblasts embedded within the mineralized matrix of bone, represent the majority of bone cells, accounting for over 90% of all cells found within the matrix or on the surfaces of the bone [189]. In the last decade, thorough research has established that osteocytes play a pivotal role in unraveling how the skeleton responds to PTH. The PTH1R receptor found in cells that express the Dentin matrix acidic phosphoprotein 1 (DMP1)-8kb gene promoter is necessary to keep the basic levels of bone resorption normal. However, this receptor is not crucial for the increased bone breakdown that occurs due to cPTH treatment; this suggests alternative pathways are at play in the heightened catabolic response to cPTH [190] (Figure 3).

a.The downstream regulatory network of PTH function in osteocytes

PKA-SIK pathway

Salt-inducible kinases (SIKs) have been identified as crucial effectors within the PTH-PKA signaling pathway in osteocytes. In the absence of PTH signaling, SIK2 tonically phosphorylates its substrates HDAC4/5 and CRTC2, leading to their cytoplasmic retention via binding to 14-3-3 chaperones. Upon activation by PTH, intracellular signaling leads to the activation of PKA, which subsequently phosphorylates and inactivates SIK2. When SIK2 is active and unphosphorylated, it phosphorylates histone deacetylases HDAC4 and HDAC5. However, when PTH-mediated signaling leads to the phosphorylation and inactivation of SIK2, this allows HDAC4 and HDAC5 to translocate into the nucleus. Within the nucleus, HDAC4 and HDAC5 act to inhibit the function of MEF2C [191], a transcription factor that normally drives the expression of the Sost gene, which is known to encode a protein that inhibits bone formation. Simultaneously, the inactivated SIK2 reduces the phosphorylation of CREB-regulated transcription coactivator 2 (CRTC2) and leads to its dephosphorylation by an unknown phosphatase, therefore promoting its nuclear translocation. Once in the nucleus, CRTC2 enhances the activity of CREB, a transcription factor that, among other things, increases the expression of the RANKL gene [192,193]. Thus, the PTH-PKA-SIK2 signaling axis has a dual role in bone metabolism: it indirectly promotes bone formation by reducing Sost expression and simultaneously enhances bone resorption through increased RANKL expression. SIK2 inhibition represents an intracellular mechanism to ensure that PTH signaling stimulates both bone formation and bone resorption.

Since SIK2 suppression leads to the regulation of key PTH target genes in osteocytes, not surprisingly, small molecular SIK2 inhibitors such as YKL05099 mimic many of the effects of PTH in osteocytes and following in vivo administration [191]. Overall, these observations indicate that SIK inhibition is a key signaling mechanism used by PTH to accomplish its physiologic effects in osteocytes.

Wnt pathway

PTH is known to activate the canonical Wnt signaling pathway and boost the production of WISP2 specifically in osteocytes, leading to an enhancement of the mineralization process [194]. In addition, LRP6 serves as a co-receptor for various G protein-coupled seven-transmembrane receptors, like Wnt receptor FZD and PTH1R, and plays a role in the production of cAMP. Sclerostin and Dickkopf WNT signaling pathway inhibitor 1 (DKK1), both antagonists in the Wnt signaling pathway, can bind to LRP6 and increase the availability of LRP6 to interact with other molecules but also inhibit the PTH-stimulated production of cAMP, thereby impacting PTH’s signaling efficiency [195]. On the other hand, PTH inhibits sclerostin production in osteocytes, which in turn increases the availability of LRP6 for PTH signaling. This suppression constitutes a positive feedback mechanism, whereby the decrease in sclerostin by PTH allows for greater LRP6-mediated signaling for both the PTH1RR and Wnt/β-catenin pathways, which is instrumental in promoting bone formation and mineral density [191].

Moreover, the N-cadherin gene (CDH2) is pivotal in regulating the interaction between LRP6 and PTHR1 in response to PTH. The deletion of CDH2 is linked with an increase in PTH-induced PKA signaling, leading to PKA-dependent phosphorylation of β-catenin at the C-terminus, a post-translational modification that enhances β-catenin’s transcriptional activity, which in turn enhances the bone anabolic effects of PTH [196].

Besides, sclerostin is a well-established inhibitor of the Wnt signaling pathway, which is vital for bone formation. Normally, the administration of PTH leads to a reduction in sclerostin levels in osteocytes, thereby facilitating Wnt signaling and promoting bone formation [197]. This is highlighted by the observation that in animals with Ocy-PTH1R^cKO^, PTH fails to suppress the expression of Sost and its protein product sclerostin [198]. It has been suggested that the downregulation of Sost, leading to the activation of the Wnt pathway, is necessary for the stimulating impact of PTH1R signaling in osteocytes and also in DMP1-caPTHR1 mice [199]. Mechanistically, PTH induces the nuclear accumulation of histone deacetylases HDAC4 and HDAC5, which result from the inhibition of SIK2. The nuclear translocation of HDAC4/5 leads to the downregulation of myocyte enhancer factor 2 (MEF2), a transcription factor that typically upregulates Sost expression [129,200,201]. In contrast, the suppression of Sost and MEF2 by PTH, along with the changes in HDAC levels, is markedly less effective in mice that lack the LRP6 receptor [200,201,202]. This finding indicates the essential role of LRP6 in facilitating the PTH-driven signaling that leads to the suppression of sclerostin, highlighting LRP6’s importance in PTH’s bone anabolic effects. Moreover, CDH2 has been identified as a protein that negatively influences the interaction between LRP6 and the PTH1R receptor, which is a necessary step for PTH to exert its effects [203]. By modulating the LRP6-PTHR1 interaction, N-cadherin indirectly mediates the suppressive action of PTH on Sost expression.

In addition to SIK2-HDAC4/5 and LRP6-Wnt regulation, Periostin (Postn) [204,205] and Tgif1 [206] have also been considered to be the PTH1R-signaling targets and involved in the PTH-inhibited expression of Sost. Recombinant Periostin has been found to suppress Sost expression, a process that is mediated through the integrin αVβ3 receptor. Conversely, using a Periostin-blocking antibody has been shown to prevent the inhibition of MEF2C and Sost by PTH [205]. Moreover, upon PTH treatment, Tgif1 is upregulated via the PKA–pCREB and prompted to disengage from the Sost promoter, which also leads to a downregulation of MEF2C expression, resulting in decreased sclerostin production [206].

Notch pathway

Recent studies have investigated the relationship between PTH treatment and the Notch signaling pathway in osteocytes, yielding contrasting findings. In the beginning, it was reported that PTH treatment stimulates osteoblastic Jagged1 mainly through the Adenylyl cyclase (AC)/PKA signaling pathway downstream of the PTH1R, and subsequently promotes osteogenesis in osteoblastic cells. However, increasing studies have demonstrated that a brief period (6 h) of PTH administration downregulates the expression of Notch pathway genes, hinders the DNA binding of the key Notch transcription factor Rbpjκ, and diminishes Notch signaling like Notch2, an inhibitor on osteogenic function, in osteocytes, which seems to be crucial for PTH’s anabolic effects [207,208,209]. A possible explanation is that Jagged1 induction is not sufficient to overcome the inhibitory effect of PTH on Rbpjκ-mediated signaling that occurs primarily in the osteoblastic lineage but might not take place in the bone marrow niche.

In contrast, other studies have reported that prolonged iPTH treatment boosts the expression of Notch pathway components like Notch1 in osteocytes, contributing to a reduced bone resorption and increased bone formation [210]. The activation of Notch1 promotes the expression of OPG and also inhibits Sost in osteocytes, which increases bone formation [211]. The length of the treatment (30 days vs. hours) and the model used (mice vs. bone organ cultures) may account for the differences between these two groups of studies.

#### 3.2.3. The Downstream Regulatory Networks of PTH That Are Anabolic in T Cells 

Earlier research has indicated that T cells facilitate PTH’s role in bone remodeling, and more recent studies have shed light on the mechanisms involved. PTH may promote bone formation by inducing T cells to release Wnt ligands, thereby stimulating the canonical Wnt pathway. Both intermittent and continuous PTH administration are known to increase the production of Wnt10b by T cells [212,213], leading to an increase in the generation of new osteoblasts and a decrease in osteoblast apoptosis. Moreover, the expression of the T cell surface receptor CD40L facilitates iPTH’s anabolic actions by triggering CD40 signaling pathways in SSCs and enhances the production of Wnt10b in T cells. However, in mice lacking CD40L—either universally or specifically in T cells—iPTH cannot effectively induce SSC proliferation, osteogenic differentiation, or activate Wnt signaling [214,215].

Summary: Multiple cellular processes, induced by PTH administration, are anabolic. These signaling networks are spatially and temporally dependent.

(2)Cellular actions of PTH as a catabolic modulator

The principal catabolic effect of PTH is increased bone resorption, a critical process for regulating calcium balance in the body. PTH1R receptors are present on osteoclasts as noted in the study by Liu et al. in 2016, which showed that continuous exposure to PTH (cPTH) can enhance the bone resorption activity of osteoclasts. This enhancement is facilitated by influencing the expression of subunits of the vacuolar-type H+-ATPase (V-ATPase), leading to increased intracellular acidification and heightened V-ATPase activity [216]. Notwithstanding, the prevailing view remains that cPTH’s direct effects on osteoclasts are limited. cPTH has been shown to impede the differentiation of osteoblasts and reduce the formation of mineralized nodules [217]. In addition to this recognized action, most previous laboratory experiments reveal that cells resembling osteoclasts do not exhibit heightened activity when exposed to PTH. This suggests that PTH prompts bone resorption not directly, but rather through stimulating osteoblasts, which in turn influence osteoclasts [100]. Research has demonstrated that in animals lacking the PTH1R receptor in osteocytes (Ocy-PTH1RKO), PTH was unable to inhibit the production of Sost or to stimulate the expression of RANKL compared to normal controls. Additionally, the process of osteoclastogenesis, which is the formation of osteoclasts, was notably hindered in Ocy-PTH1RKO models when treated with PTH in a controlled environment. This suggests that osteocytes are pivotal in managing the formation of osteoclasts through pathways involving the PTH1R receptor [218] (Figure 4).

#### 3.2.4. The Downstream Regulatory Networks of PTH That Are Catabolic in Both Osteoblasts and Osteocytes

PKA and PKC pathwaysPKA pathway

cPTH treatment can prompt osteoblasts to produce MCP-1 via the cAMP-PKA pathway. This expression leads to the recruitment and formation of osteoclast precursors and their subsequent fusion into mature, multinucleated osteoclasts that are essential for bone resorption [140,219]. On the flip side, mice lacking MCP-1 display resistance to bone loss typically induced by cPTH, which includes both cortical and trabecular bone regions [220].

E4-binding protein 4 (E4BP4) has been pinpointed as a potential mediator that plays a part in the catabolic activities of PTH and PTHrP on bone. It was one of the cAMP-PKA pathway targets in response to PTH, and it exerts its effects by downregulating the expression of pivotal genes such as phosphate regulating endopeptidase X-Linked (PHEX), RUNX2, and OSX—each of which is integral to the differentiation and functional activity of osteoblasts [221].

PKC pathway

Enhanced signaling through the Gαq/11 pathway has been found to have an inhibitory effect on the osteoanabolic actions of PTH [222], and the osteoanabolic action of PTH is actually enhanced when there is a deficiency in the Gα(q) signaling pathway within cells, a process that seems to be autonomous to the cells themselves. This enhancement is associated with the movement of PKCδ to the cell membrane [223].

Furthermore, the serine phosphorylation at position 79 within the 63–84 domain of CREB-binding protein/p300-interacting transactivator with Asp/Glu-rich C-terminal domain 1 (CITED1) is linked with the activation of PKC. This phosphorylation is essential for the translocation of CITED1 to the nucleus following stimulation by PTH. Once in the nucleus, CITED1 acts as a transcriptional co-factor that suppresses the osteoblastogenesis and mineralization, thereby diminishing the osteoanabolic effects of PTH [224].

OPG-RANKL

Research has revealed that PTH1R can activate beta-catenin signaling pathways by directly recruiting Dvl, independently of Wnt or the co-receptors LRP5/6. However, the β-catenin signaling pathway can be activated in distinct ways by FZD and PTH1R, which may lead to convergent or divergent outcomes in terms of osteoblast differentiation and osteoclastogenesis. In the case of FZD, a protein kinase A-dependent phosphorylation of β-catenin and glycogen synthase kinase 3 beta (GSK3β) in response to iPTH will induce the upregulation of OPG and leads to the inhibition of osteoclastogenesis, whereas the activation of the PTH1R in response to cPTH will increase osteoclastogenesis via the PKA-dependent activation of CREB [225].

On the other hand, cPTH has been shown to elevate the expression of RANKL in isolated murine osteocytes, resulting in both the recruitment and activation of osteoclasts, which thereby enhances bone resorption [100,226]. Studies involving transgenic mice with DMP1-caPTHR1 demonstrate that activation of this signaling within osteocytes alone is enough to increase RANKL expression and leads to enhanced bone resorption [227], indicating the potency and specificity of PTH receptor signaling in osteocytes for its crucial role in bone resorption and catabolic action. Moreover, in conditions lacking 17β-estradiol, PTH has been observed to increase the OPG/RANKL ratio. Conversely, when estradiol is present, the OPG/RANKL ratio is decreased, suggesting a promotion of bone resorption [228]. The results infer that estradiol is a key modulator of PTH and significantly influences the mRNA expression levels of OPG and RANKL in osteoblastic cell lines [229].

Mechanically, PTH regulates the expression of RANKL through several intricate cellular mechanisms involving the MEF family, particularly MEF2C. PTH activates PKA, which phosphorylates serine641 residues of the MEFC, thereby increasing the expression level of RANKL [230]. Additionally, PTH can initiate the ubiquitination and subsequent degradation of HDAC4 through the PKA-SMAD ubiquitination regulatory factor 2 (SMURF2) pathway [231,232]. The degradation of HDAC4 liberates MEF2C, allowing it to transactivate the RANKL promoter and thus enhance RANKL expression. Besides, MEF not only has a role in suppressing osteoblast differentiation by inhibiting the activity of RUNX2 but also is known to upregulate the expression of male abnormal gene family 21 (MAB21), which has been suggested as a potential transcriptional repressor of osteoblast differentiation. Nevertheless, it has been observed that the transcriptional effects on Tnfsf11 expression alone do not typically lead to the release of soluble RANKL (sRANKL), indicating PTH may exhibit a unique activity in promoting the proteolytic processing of sRANKL [233].

In additional to RANKL itself, pre-osteoclasts can secrete serum amyloid A3 (SAA3) in response to RANKL stimulation, and in turn, act to inhibit PTH-stimulated PKA signaling and osteoblast differentiation through the Gαi/o subunit [234]. Similarly, cPTH has been found to increase the expression of MMP14 in osteocytes. In addition, the enzymatic activity of MMP14 leads to the production of a soluble form of sRANKL [235]. Interestingly, MMP14 not only contributes to the production of sRANKL but also regulates the activity of PTH1R itself. MMP14 can cleave PTH1R, particularly at amino acid 61, inhibiting its signaling activity. This cleavage by MMP14 promotes the degradation of PTH1R, serving as a novel mechanism for modulating the stability and signaling efficacy of this G protein-coupled receptor (GPCR), thus attenuating PTH signaling [236]. Additionally, cPTH treatment may also elevate the expression of CCL2, a chemokine that works alongside RANKL to facilitate the recruitment of monocytes, which then fuse to form mature, bone-resorbing osteoclasts [219].

However, certain mechanisms have been identified that limit RANKL production, facilitating the anabolic effects of PTH. For instance, studies have shown that in osteoblasts lacking the Oncostatin M receptor, the RANKL induction following PTH treatment remains elevated for an extended period. Conversely, when the Oncostatin M receptor (OSMR) is present and activated by PTH, it curtails the duration of RANKL production in cells of the osteoblast lineage [237]. Another regulatory mechanism involves ephrinB2, whose expression in osteoblasts is induced by PTH. The interaction between ephrinB2 and its receptor EphB4 within the osteoblast lineage is essential for the later stages of osteoblast and osteocyte differentiation. Moreover, this signaling pathway limits the capacity of osteoblasts to support the formation of osteoclasts, in part by constraining RANKL production [238].

PTH1R endocytosis

Vacuolar protein sorting 35 (VPS35), a major component of retromer, has been reported to be another key regulator of PTH1R endocytosis in addition to β-arrestin. Besides PTH1R, VPS35 also interacts with protein phosphatase 1 regulatory subunit 14C (PPP1R14C), which is an inhibitory subunit of PP1 phosphatase. Through its interaction with PPP1R14C, the VPS35/retromer complex may play a role in terminating the endosomal signaling of PTH. This is achieved by reducing the binding of PPP1R14C to the PTH1R, which in turn releases the inhibitory effect of PPP1R14C on PP1 phosphatase. Consequently, this action prevents the PP1-driven dephosphorylation of endosomal signaling proteins and effectively enhances the catabolic response of cPTH [239].

#### 3.2.5. The Downstream Regulatory Networks of PTH That Are Catabolic in T Cells

cPTH administration leads to the proliferation of Th17 cells and an elevation in the inflammatory cytokine Interleukin-17A (IL-17A) in both mice and humans. Produced by Th17 cells, IL-17c serves as an “upstream cytokine”, amplifying the response of osteoblasts and osteocytes to PTH, which in turn increases the secretion of RANKL [240]. Furthermore, IL-17A increases the levels of N-cadherin, which interferes with the interaction between PTH1R and LRP-6, consequently impeding Wnt signaling and bone formation [241]. Mechanistically, the production of tumor necrosis factor alpha (TNFα) by conventional T cells stimulated by cPTH, promotes the formation of IL-17A-producing Th17 cells through the TNF receptor 1 signaling in CD4(+) cells. Additionally, cPTH makes naive CD4(+) cells more responsive to TNFα through the activation of GαS/cAMP/Ca(2+) signaling pathways.

Conversely, the deletion of GαS specifically in CD4(+) T cells [242], the inhibition of IL-17RA signaling, or the application of anti-IL-17A antibodies [241,243] in osteoblasts/osteocytes, blunts the ability of cPTH to induce RANKL production by these cells and prevents bone loss.

Summary: PTH can act as a catabolic hormone in the skeletal milieu.

### 3.3. Adipocytes and PTH

Consistent with PTH’s role in inhibiting the development of fat cells from SSCs, administration of PTH has been shown to significantly decrease the number of adipocytes and the expression of adipogenic markers such as C/EBPβ, PPARγ, and ZFP467 in BM [32,244]. PTH also counteracts the increase in marrow adiposity observed in OVX mice [31], and it reduces the heightened adipocyte density and size that occurs in mice subjected to caloric restriction [245].

On the molecular level, PTH promotes the release of fatty acids from adipocytes and their subsequent β-oxidation in osteoblasts [183,245]. This suggests that the anabolic effects of PTH on bone formation may involve a coordinated signaling between the osteogenic lineage and BM adipocytes. In this process, a lipolytic response in the adipocytes releases fatty acids, which are then utilized by osteoblasts as an energy source through β-oxidation during the bone formation process.

### 3.4. ECs and PTH

Angiogenesis plays an important role in bone healing. Although there is no difference regarding the cell proliferation or cell sentence in ECs that respond to PTH [87], numerous studies have established that PTH not only stimulates bone formation and resorption but also enhances angiogenesis and vasodilation [2,246]. On one hand, PTH has been shown to promote the formation of small blood vessels (diameter ≤ 50μm) and significantly increase the expression of the endothelial cell marker CD31 in bone calluses and trabecular bone [50]. Additionally, PTH treatment boosts the expression of angiogenic factors like VEGF, stromal cell-derived factor 1 (SDF-1), and brain-derived neurotrophic factor (BDNF) in SSCs, which in turn enhances angiogenesis in ECs [247]. Conversely, the absence of endogenous PTH can lead to reduced VEGF expression in SSCs due to downregulation of the PKA/pAKT/HIF1α/VEGF pathway, impacting angiogenesis and bone formation [248]. On the other hand, PTH enhances vasodilation in the femoral artery endothelium through increased nitric oxide (NO) production, partially mediated by VEGF signaling [249,250]. iPTH treatment also increases microvessel size and bone perfusion, which stimulates bone formation and mitigates bone perfusion and vessel density reduction caused by OVX [251]. Conversely, the inhibition of endothelial nitric oxide synthase (eNOS) or the PKA pathway can diminish PTH-induced vasodilation and its proliferative anabolic effects [252,253]. Collectively, these findings suggest that PTH’s vasodilatory effect is mediated by the activation of PTH1R on endothelial cells, leading to an increase in NO and VEGF signaling.

Recently, Kusumbe et al. [26] identified a specific subset of ECs characterized by the high expression of CD31 and endomucin (CD31hi/Endmhi), termed type H cells, which are closely linked to sites of bone formation. Interestingly, the area occupied by type H vessels in the BM and transition zone remains unchanged with iPTH treatment when compared to controls. In contrast, type H vessel area and density in cortical bone were significantly increased in the iPTH-treated group. Mechanically, iPTH treatment reduces the coverage of leptin receptor-positive (LepR(+)) cells on transitional vessels specifically, while the number of LepR(+) cells not associated with vessels rises in the endocortical area. These observations suggest that iPTH selectively modifies transitional vessels and may encourage the mobilization of LepR(+) cells away from these vessels near the bone surface [254]. Taken together, the anabolic effects of iPTH on bone could be partially attributed to this vascular remodeling, which leads to the expansion of type H cells accompanied by more pericytes, which have the potential to differentiate into osteoblasts.

Moreover, PTHrP has been found to stimulate the expression of both the protein and mRNA of MCP-1 in human BM ECs through mechanisms dependent on the binding elements of C/EBPβ) and NF-κB. This process, in turn, promotes the migration and infiltration of monocytes and enhances the differentiation of macrophages [255].

### 3.5. PTHrP as the Therapeutic Target for Skeletal Metastasis of Unknown Primary (SMUP)

Skeletal metastases of unknown primary (SMUP) represent enigmatic rare metastatic tumor entities without anatomic primary sites identified. Mechanically, the process of cancer spreading to the bone is the homing to the marrow microenvironment throughout the bloodstream of the tumor cells, via the neo-angiogenesis process, through permissive bone marrow endothelial cells [256,257]. Recently, a number of molecular actors have been considered as elements that drive the neoplastic cells to the bone environment, including adhesion molecule systems such as CXCL12/CXCR4 (chemokine ligand 12/chemokine receptor 4) [258], junctional adhesion molecules in osteotropic tumors [259,260], focal adhesion kinases [261,262], and vascular and immune–microenvironment interactions [263,264]. Once cancer cells reach the bone marrow level, they cross the wall of sinusoids, invade the matrix, and, once they reach the endosteal surface, stimulate osteoclastogenic activity and proliferate with the formation of metastases with prevailing osteolytic or osteoblastic development. These extremes cover the spectrum of neoplastic bone remodeling [265].

Numerous clinical trials are currently exploring various molecules that impact the progression of bone metastases in solid tumors, either directly or indirectly. Some of these molecules specifically target the bone resorption process by focusing on certain bone cells like osteoclasts, osteoblasts, osteocytes, or by influencing molecular pathways that regulate these cells’ functions [266]. Key molecules under investigation include denosumab [267], which inhibits the endothelin 1 receptor, drugs that disrupt the Wnt/DKK1 pathway [268], and Src inhibitors [269]. However, the primary focus of these drugs so far has been their ability to halt the destruction of the bone matrix caused by tumor cells. These treatments aim to intervene in the complex interactions between cancer progression and bone health, potentially offering new avenues for managing bone metastases in cancer patients. Furthermore, emerging research is lending support to the theory that PTHrP plays a significant role in influencing the bone microenvironment. It has emerged as a key player in various cancers, particularly in their skeleton metastasis [270,271,272]. It affects cells within the bone microenvironment, exerting proliferative and pro-survival effects that prepare both the ‘seed’ (the cancer cells) and the ‘soil’ (the bone environment) for the establishment and growth of metastatic lesions [273]. This new understanding of PTHrP’s role highlights the complex dynamics between cancer progression and bone biology, opening up potential avenues for targeted therapies in oncology.

## 4. Conclusions

The initial understanding of PTH’s role in the skeleton was primarily as a promoter of bone resorption, yet it is now acknowledged that PTH can indeed stimulate bone formation. The ultimate impact of PTH on bone mass, whether it is anabolic or catabolic, hinges on the balance between these processes. The determination of PTH’s effect is thus contingent on which of these actions, bone formation or resorption, is predominantly influenced by its activity on mesenchymal cells within the BM niche. In brief, continuous exposure to high levels of PTH is typically linked with bone degradation, while periodic exposure to low doses has been found to promote bone growth.

When administered intermittently, PTH stimulates the transformation of SSCs into osteoblasts, enhances the growth and maturation of osteoblasts or osteocytes, and significantly fosters bone formation. Mechanistically, iPTH bonds to PTH1R and activates Gαs in SSCs, osteoblasts, and osteocytes, and further stimulates the production of cAMP and initiates the PKA cascade reaction. PKA can promote the phosphoration of several downstream targets including α-NAC, CREB, β-catenin, and SIK2, inducing the osteogenic gene transcription, inhibiting gene expression of Sost, and thereby increasing osteogenesis and cell migration. MEF2C was also considered to be a key effector downstream of PTH1R signaling that mediates the gene expression of Sost and MMP13. Besides, treatment with iPTH can enhance the levels of IGF1, Amphiregulin, Wnt10b, and BMPs leading to the activation of their respective receptors, IGF1R, EGFR, Wnt Frizzled, and BMPR, thereby initiating their downstream IGFR-mTOR-AKT, EGFR-p38, Wnt/β-catenin and BMPR-SMAD1/5/8 signaling pathways. PTH1R is known for its signaling at the cell surface, but interestingly, it also continues to signal after being internalized into the cell through a process called endocytosis. Once internalized, PTH1R not only extends the activation of PKA but also activates the p38 and ERK-MAPK pathways, thereby contributing to cell survival. Moreover, PTH1R has been observed to influence cell metabolism, encompassing the enhancement of glycolysis, modulation of ER stress, and reduction in ROS production. These activities collectively contribute to increased osteogenesis and the inhibition of apoptosis in osteoblasts.

With continuous administration, PTH indirectly influences osteoclast formation through its influence on osteoblasts and osteocytes, thereby encouraging their growth, development and cell fusion, which leads to increased bone resorption. Upon the binding of PTH1R by cPTH, a cascade of intracellular events is triggered where PKA phosphorylates CREB and inhibits HDAC2, thereby enhancing the transcriptional activity of MEF2C, leading to increased gene expression of RANKL and MAB21. Additionally, cPTH stimulates osteoblasts and osteocytes to produce and secrete various factors like MCP-1, CCL2, MMP14, and SAA3. All these secreted factors play a crucial role in promoting the differentiation and cell fusion of osteoclast precursors.

The mechanism by which PTH regulates bone involves not only cells of osteoblastic lineage but also T cells from the immune system. This interaction underscores the complexity of PTH’s role in bone regulation, suggesting that its influence extends beyond the skeletal system and into the immune system. There is a reciprocal regulatory relationship between immune and bone cells, indicating a significant interplay between these two systems in the context of PTH’s function. In reaction to PTH treatment, T cells are capable of producing either Wnt10b or Th17α, thereby playing a role in both the anabolic and catabolic actions of PTH within the BM. Emerging research has also revealed that B cells can produce RANKL, which subsequently increases osteoclast formation [274]. Given this information, it can be reasonably inferred that B cells are also involved in the bone remodeling mediated by PTH. In addition to normal cells, it has been reported that tumor cells can secrete PTHrP, which stimulates bone resorption via increasing RANKL expression in osteoblastic cells [275]. Besides, the produced PTHrP by tumor cells could also act on osteoblastic cells to increase the expression of Jagged1, which may promote angiogenesis by activating Notch signaling on HSCs and support the tumor cell niche in bone [276]. This interaction distinctively enhances signals beneficial for tumor localization in bone. While PTHrP is recognized as a harmful factor in tumor development within metastatic lesions, more research is required to accurately determine the relationship between PTHrP and tumor progression in humans [275,277].

Taken together, this review compiles a broad array of research highlighting the significance of PTH1R signaling in the regulation of mesenchymal cells within the BM niche. While the therapeutic benefits of PTH-related treatments are clear, there remains a need for ongoing research to fully elucidate the mechanisms of action and to optimize therapeutic strategies for individual patients. For the future studies, on the one hand, a more detailed understanding of how PTH interacts with its receptors and the downstream effects of this interaction is necessary, which is crucial for developing targeted therapies for diseases related to PTH dysregulation. On the other hand, increasing studies are being conducted to explore the intricate relationships between PTH and various aspects of cellular and physiological function. For instance, the role of PTH in carbohydrate and lipid metabolism in the liver, and the specific effects of different regions of the PTH molecule on skeletal development and survival are areas of active investigation. These research endeavors are expanding our understanding of the diverse roles that PTH plays in the body beyond its traditional association with calcium homeostasis. The exploration of these and other areas reflects a growing recognition of the complexity of PTH’s function and the underlying mechanisms. In this review, we not only underscore the critical function of PTH1R in bone remodeling and bone physiology but also elucidate the therapeutic potentials of PTH and PTHrP in the management of osteoporosis.

## Figures and Tables

**Figure 1 cells-13-00406-f001:**
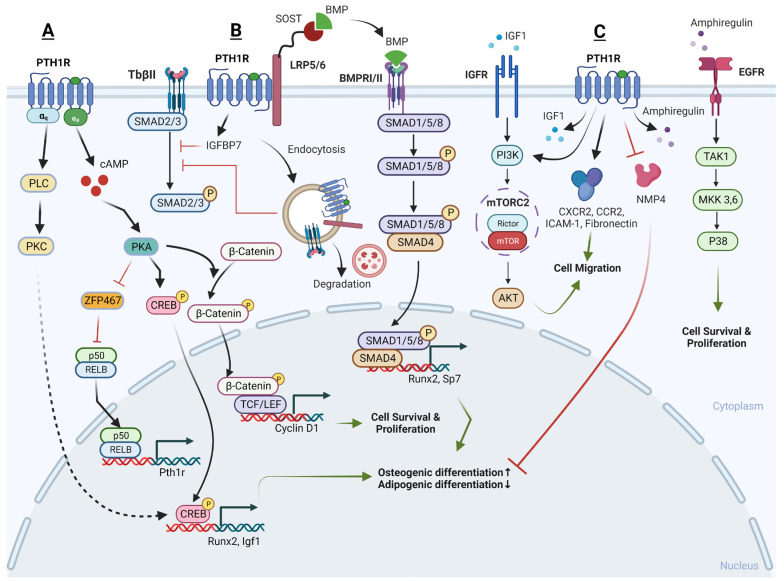
Downstream pathways triggered by PTH1R signaling in SSCs. (**A**) Activation of PKA via Gαs and PKC via Gαq by the PTH-bound PTH1R, with PKA phosphorylating CREB and β-catenin to activate their downstream targets. (**B**) Interaction of PTH1R with receptors like TbβII, LRP5/6, and BMPRI/II. PTH1R activation may induce the endocytosis of TbβII and LRP5/6, leading to SMAD2/3 inhibition and SMAD1/5/8 pathway activation. (**C**) Elevation in IGF and Amphiregulin expression by activated PTH1R, triggering IGF1R and EGFR, which then activate the AKT and p38 pathways, enhancing cell survival and proliferation.

**Figure 2 cells-13-00406-f002:**
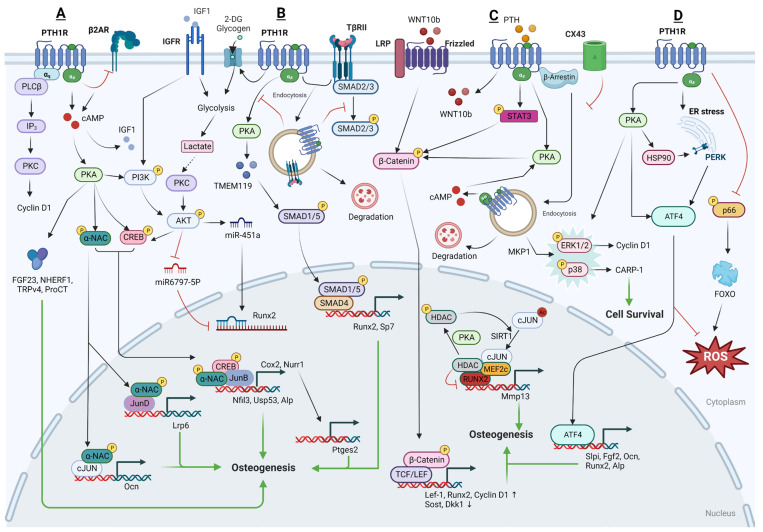
Schematic model showing the process by which PTH induces anabolic activities in osteoblasts. (**A**) The PTH-bound PTH1R activates PKA via Gαs and PKC via Gαq, with PKA phosphorylating α-NAC and CREB, leading to the activation of their downstream target transcription. (**B**) Activation of PTH1R enhances glycogen transport and boosts intracellular glycolysis, resulting in increased lactate production and activation of the PKC-AKT pathway. Additionally, PTH1R interacts with TbβII and may induce endocytosis of TbβII upon PTH1R activation, inhibiting the SMAD2/3 pathway. (**C**) PTH1R activation can lead to the phosphorylation of PKA and STAT3, thereby activating the Wnt/β-catenin pathway. The PTH-bound form of PTH1R recruits β-arrestin and decouples from the G protein, followed by internalization. Inside the endosome, the PTH-PTH1R-β-arrestin complex activates the MAPK pathway and ERK 1/2 pathway. This complex can also re-recruit the G protein, inducing a prolonged cAMP signal. (**D**) PTH1R signaling plays a role in regulating ER stress and ROS production.

**Figure 3 cells-13-00406-f003:**
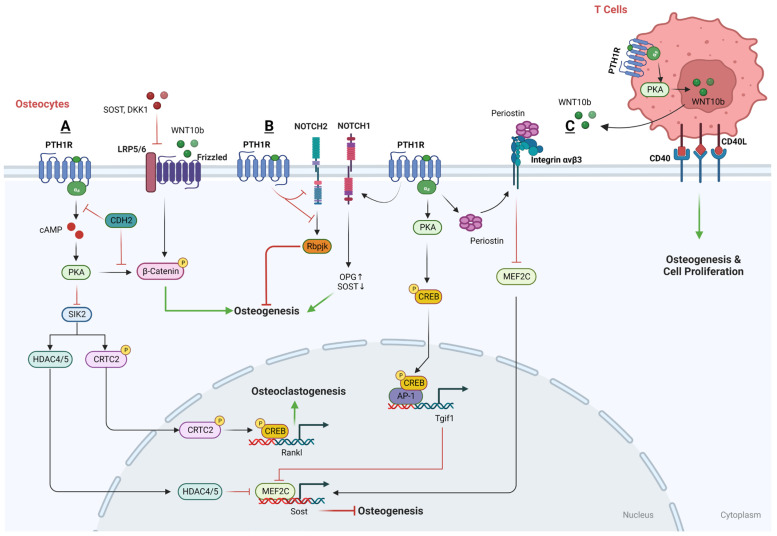
Schematic model showing the process by which PTH induces anabolic activities in osteocytes. (**A**) The PTH-bound form of PTH1R activates PKA via Gαs. PKA phosphorylates SIK2, inhibiting its ability to phosphorylate HDAC4/5. The dephosphorylated HDAC moves into the nucleus, where it inhibits the expression of Sost by repressing MEF2C. PKA also phosphorylates β-catenin, enhancing osteogenesis. (**B**) iPTH suppresses NOTCH2 signaling, which inhibits osteogenesis by increasing the DNA binding of Rbpjk. Conversely, iPTH promotes NOTCH1 signaling and increases OPG expression, favoring bone formation. (**C**) The expression of the T cell surface receptor CD40L supports the anabolic actions of iPTH by initiating CD40 signaling pathways in SSCs. This interaction also boosts the production of Wnt10b in T cells, further promoting bone formation.

**Figure 4 cells-13-00406-f004:**
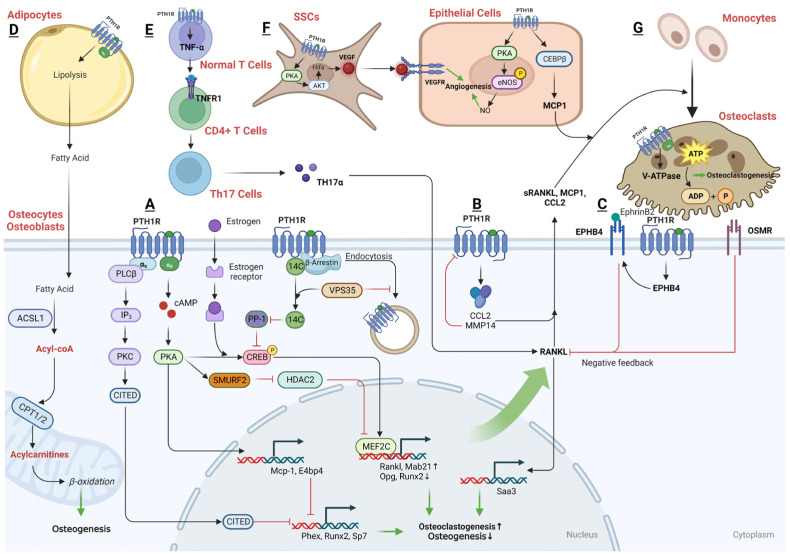
Schematic model showing the process by which PTH induces catabolic activities in the BM niche. (**A**) Upon the binding of PTH1R by cPTH, a cascade of intracellular events is triggered where PKA phosphorylates CREB and inhibits HDAC2, thereby enhancing the transcriptional activity of MEF2C, leading to the increased gene expression of RANKL and MAB21. (**B**) cPTH stimulates osteoblasts and osteocytes to produce and secrete various factors like MCP-1, CCL2, MMP14, and SAA, which play a crucial role in promoting the differentiation and cell fusion of osteoclast precursors. (**C**) OSMR and EPHB4 signaling have been found to reduce the expression of RANKL downstream of cPTH, thereby limiting the bone resorption typically induced by PTH. (**D**) PTH promotes the release of fatty acids from adipocytes and their subsequent β-oxidation in osteoblasts. (**E**) The production of TNFα by conventional T cells, stimulated by cPTH, enhances the differentiation of IL-17A-producing Th17 cells. As a result, there is an increase in RANKL expression in cells of the osteoblastic lineage. (**F**) Upon treatment with PTH, the expression of VEGF is upregulated in SSCs through the PKA/AKT/Hif1a pathway. This upregulation leads to the activation of the VEGFR pathway in ECs. Simultaneously, the activation of PTH1R in ECs promotes the production of NO via the PKA pathway, which subsequently enhances angiogenesis. (**G**) cPTH enhances the bone resorption activity of osteoclasts by directly influencing the expression of V-ATPase subunits, intracellular acidification, and V-ATPase activity.

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
