# Peer review of "PTH and the Regulation of Mesenchymal Cells within the Bone Marrow Niche"

_cells, 2024, doi:10.3390/cells13050406_

Round 1

Reviewer 1 Report

Comments and Suggestions for Authors

This review comprehensively summarizes the current state-of-knowledge on the regulatory role of parathyroid hormone (PTH), with a particular focus on various types of bone cells, such as skeletal stem cells, osteoblastic lineage cells, osteocytes and T cells. The review discussed the complex interplay of signaling pathways and regulatory factors in the anabolic and catabolic actions of PTH. 

The reviewer suggests that the structure and logical sequence of section 3 should better be rearranged. For instance, subtitles or a paragraph focusing on “The downstream regulatory network of PTH function in SSCs” can be included in line 291 after 3.1.4 Angiogenesis. It is the same in the osteoblast and osteocytes section. Furthermore, 3.2 osteoblasts and osteocytes section can be reorganized. For example, PTH1R endocytosis, cell metabolic regulation, microRNA can be separated from the signaling pathways. Regarding osteocytes part, T cells section can also be isolated from the summary of pathways. The author can also consider summarizing another part focusing on PTH1R endocytosis which consists of different cell types. In conclusion part, the authors can include research advances and novel topics to shed light on the future studies of PTH’s function and the underlying mechanisms.

Author Response

This review comprehensively summarizes the current state-of-knowledge on the regulatory role of parathyroid hormone (PTH), with a particular focus on various types of bone cells, such as skeletal stem cells, osteoblastic lineage cells, osteocytes, and T cells. The review discussed the complex interplay of signaling pathways and regulatory factors in the anabolic and catabolic actions of PTH. 

The reviewer suggests that the structure and logical sequence of section 3 should better be rearranged. For instance, subtitles or a paragraph focusing on “The downstream regulatory network of PTH function in SSCs” can be included in line 291 after 3.1.4 Angiogenesis. It is the same in the osteoblast and osteocytes section.

Response: Thank you so much for your suggestions, we added related subtitles as follows in section 3:

3.1.5 The downstream regulatory network of PTH function in SSCs

The downstream regulatory network of PTH function in Osteoblasts

The downstream regulatory network of PTH function in Osteocytes

The downstream regulatory network of PTH catabolic function

Furthermore, 3.2 osteoblasts and osteocytes section can be reorganized. For example, PTH1R endocytosis, cell metabolic regulation, and microRNA can be separated from the signaling pathways. Regarding the osteocytes part, T cells section can also be isolated from the summary of pathways. The author can also consider summarizing another part focusing on PTH1R endocytosis which consists of different cell types. 

Response: Thank you for your valuable suggestions. We moved “PTH1R endocytosis, cell metabolic regulation, microRNA” to a new part “Cellular Processes and other Regulatory Mechanisms” in section 3.2 after “The downstream regulatory network of PTH function in Osteoblasts”. We added “(3) Osteocytes-T cells cross talk” part in section 3.2 to discuss about osteocytes-t cell cross talk.

In the conclusion part, the authors can include research advances and novel topics to shed light on future studies of PTH’s function and the underlying mechanisms.

Response: That’s a good point, we added more discussion about research advances and novel topics that can be studied in future studies in the conclusion part as follows:

“…For future studies, on the one hand, a more detailed understanding of how PTH as a ligand interacts precisely with its receptors and the downstream effects of this interaction is necessary, which is crucial for developing targeted therapies for diseases related to PTH dysregulation. On the other hand, more studies are being conducted to explore the intricate relationships between PTH and various aspects of cellular and physiological function. For instance, the role of PTH and PTHrp in carbohydrate and lipid metabolism in the liver, and the specific effects of different regions of the PTH molecule on skeletal development and survival are areas of active investigation. These research endeavors are expanding our understanding of the diverse roles that PTH plays in the body beyond its traditional association with calcium homeostasis. The exploration of these and other areas could reflect a growing recognition of the complexity of PTH’s function and the underlying mechanisms.”

Reviewer 2 Report

Comments and Suggestions for Authors

This is an ambitious and comprehensive review aimed at describing the complex effects of PTH in the bone marrow niche. However, its content fits rather with an overall description of the multiple PTH actions within bone remodelling units on bone surfaces. Osteocytes, e.g., fully differentiated osteoblasts embedded in the mineralized bone matrix, are unlikely to be a cell component in the bone marrow niche. Indeed, the majority of the cited reports describe PTH actions in more differentiated osteoblastic cells rather than SSCs.

In my opinion, it would have been more appropriate to start describing the well characterized effects of PTH on cells involved in bone remodeling, followed by its less chracterized actions on SSCs.

The review is an update of recent findings on this topic; in fact, many previous reviews are included in References. Generally, there are frequent redundancies throughout the text, making the text hard to read on occasion. Various PTH induced mechanisms are not sufficiently discussed, mainly those associated with dual PTH actions. Specific concerns in this regard are as follows:

1. The putative role of Nestin+ SSCs in osteogenesis/adipogenesis is not defined (lines 80-3).

2. The effects of PTH on osteoblastic differentiation (section 3.1.1) are not clearly described; the study in ref. 29 was performed in a mouse fetal osteoprogenitor cell line, not in mature osteoblasts as stated.

3. The 2nd and 3rd paragraphs of section 3.1.4. deal with PTH effects on osteoprogenitors and hematopoiesis, respectively, not angiogenesis as the section title suggests. In particular, the putative impact of the latter effects on SSC osteogenic differentiation should be more clearly elaborated (lines 283-4).

4. The apparently controversial effects of iPTH on TGFbeta signaling (TGF beta pathway section), and also the interaction between PTH1R and BMP signaling, are unclear. Differences in studies of references 117 and 118 regarding the possible impact of beta adrenergic receptor on PTH anabolism are not explained (lines 530-42).

5. The lack of effect of PTH on SSC senescence (line 426) contrasts with what is mentioned previously (line 355).

6. The role of PTH-induced SIRT1 as a putative mechanism to prevent MMP13 overexpression in osteoblasts is unclear (lines 681-4).

7. The dual role of PTH through the PKA-SIK2 signaling pathway in bone remodeling (lines 781-96), as well as the true role of the Notch pathway in osteocytes as a mediator of PTH anabolism (lines 816-26), should be more elaborated.

8. The studies involving SOST should be included in the Wnt pathway section (line 797). PTH inhibits not induces SOST (line 853).

9. T cells subsection (line 859) is misplaced under the Osteocytes section (line 772).

10. RANKL acts on preosteocalsts not osteoblasts to secrete SAA3 in Choudhary et al. (ref. 232, line 956).

11. The PTH actions described in sections 3.3 (Adipocytes) and 3.4 (Endothelial cells) are unrelated to PTH catabolism in bone.

12. PTH has been shown to stimulate MCP-1 in various cell types (kidney, osteoblasts,...); thus, results described in lines 1068-71 are not really unexpected.

13. It is now known that interaction of PTH and PTHrP with different PTH1R conformations might help explain differences in signaling pathways leading to a predominant biological response (anabolic vs catabolic). This particular aspect should be included in this review.

Other points.-

- Acronymes must be described the first time mentioned.

- Reviews are abundant. Some of them might be replaced by an original paper.

- The term "non-redundant" (line 147) is not accurate, since PTHrP is the PTH counterpart in bone tissue (not "a hormone", line 210).

- The first author of an article must be followed by "et al" when referenced in the text. Figures citation should be included where corresponding.

Comments on the Quality of English Language

In general, the text seems to be carelessly written; e.g., a significant number of missing and/or redundant prepositions occur in several sentences. The text contains many redundancies and overlaps in different sections, which makes it heavy reading.

Author Response

  1. The putative role of Nestin+ SSCs in osteogenesis/adipogenesis is not defined (lines 80-3).

Response: Thank you for your correction, we revised the description about Nestin+ SSC as followed:

“Nestin, an intermediate filament, was considered to be another characteristic marker of BM SSCs,  entire mesenchymal activity and clonogenicity of pre-reported CD45− SSCs reside within the Nestin+ subset. Nestin+ SSCs are usually present in the periosteum and localize closely to blood vessels, they show properties of skeletal progenitors in vitro and can form bone in vivo. However, Nestin+ SSCs are not slow dividing nor self-renewing in vivo. NG+ positive cells were referred to as the periarteriolar cell population strongly expressing nestin. HSCs are observed to be closely adjacent to the Nestin+ cells, and Nestin+ cells may have essential role for the control of HSC quiescence and haematopoiesis”

  1. The effects of PTH on osteoblastic differentiation (section 3.1.1) are not clearly described; the study in ref. 29 was performed in a mouse fetal osteoprogenitor cell line, not in mature osteoblasts as stated.

Response: Thank you for the corrections, we deleted the wrong description about mature osteoblasts of ref. 29. We re-wrote the section 3.1.1 as followed”

“A multitude of studies have indicated that PTH can modestly stimulate the proliferation of SSCs. In addition to one particular study highlighted that PTH impedes the differentiation of cells expressing Runx2, especially in the context of matrix mineralization29, most studies revealed that PTH has a pronounced effect on osteogenic differentiation by enhancing alkaline phosphatase activity and calcium production in a dose-dependent fashion30-31. In response to iPTH treatment, LepR(+) MSPCs could differentiate into mature osteoblasts expressing Col1, and the proliferative LepR-Cre-marked lineage cells near bone tissue also increase. The expression level of osteoblast markers including Sp7 and Col1 rises, while adipocyte markers Cebpb, Pparg, and Zfp467 decline in the LepR(+)  SSCs population, indicating a shift towards osteogenic differentiation and reduced adipogenesis due to iPTH treatment35. Consistently, studies have demonstrated that PTH can have a rejuvenating effect on SSCs isolated from patients with atypical fractures, and these cells exhibited an increase in their capacity to differentiate into bone-forming cells with PTH treatment. PTH appears to enhance both the number and functional activity of osteoprogenitor SSCs33. In agreement with previous study that PTH was shown to increase markers of osteogenic differentiation and promote the migration of SSCs by upregulating the expression of SDF-132, iPTH treatment has also been observed to increase the density of circulating SSCs, characterized by markers CD73+, CD90+, CD105+, CD34-, and CD45-, in women with postmenopausal osteoporosis. And not surprisingly, iPTH treatment has shown to enhance the in vitro osteogenic differentiation potential of these cells34.”

  1. The 2nd and 3rd paragraphs of section 3.1.4. deal with PTH effects on osteoprogenitors and hematopoiesis, respectively, not angiogenesis as the section title suggests. In particular, the putative impact of the latter effects on SSC osteogenic differentiation should be more clearly elaborated (lines 283-4).

Response: We totally agree with your points, to avoid confusion, we added two section titles”3.1.5 In Vivo Models in PTH and SSCs Study and 3.1.6 Indirect effect of PTH on hematopoiesis via regulating SSCs” for the 2nd and 3rd paragraphs of section 3.1.4. As for the part that “The indirect effect on hematopoiesis has a beneficial effect on SSCs osteogenic differentiation, thereby supporting the differentiation of osteoprogenitors and contributing to the anabolic bone remodeling effects of PTH”, it would be too much to talk about the indirect impact of increasing hematopoiesis on SSCs. Therefore, we did not include that part of work here.

  1. The apparently controversial effects of iPTH on TGFbeta signaling (TGF beta pathway section), and also the interaction between PTH1R and BMP signaling, are unclear. Differences in studies of references 117 and 118 regarding the possible impact of beta adrenergic receptor on PTH anabolism are not explained (lines 530-42).

Response: Thank you for your suggestions, we revised the TGFbeta signaling section, BMP section and β2-AR section as followed:

TGF-β pathway

iPTH was found to upregulate the transcription of genes related to osteogenesis and the expression of IGFBP7 in both SSCs and pre-osteoblasts. IGFBP7 is believed to inhibit the mTOR pathway while enhancing the TGF-β/BMP and canonical Wnt signaling cascades71. However, in an osteoarthritis model with excessive active TGF-β signaling, iPTH plays a crucial role in maintaining coordinated subchondral bone remodeling as a negative feedback break. iPTH mitigates the increases in both Nestin-positive and Osterix-positive celss within the BM and induces the formation of osteoids on the surface of subchondral bone. When iPTH is initiated early, it helps attenuate abnormal bone remodeling in osteoarthritis patients by regulating the recruitment and migration of SSCs in the BM. Mechanistically, on the one hand, PTH was found to inhibit the phosphorylation level of SMAD3, a protein that can bind to the p16(ink4a) gene promoter region. This inhibition leads to a reduction in the accumulation of senescent cells and an increase in cellular proliferation of SSCs73. On the other hand, in response to PTH, PTH1R and transforming growth factor-beta receptor II (TbRII) form an endocytic complex. PTH induces the endocytosis of TbRII, leading to a decrease in TGF-β1-induced phosphorylation and the nuclear translocation of its downstream targets, SMAD2 and SMAD3, when compared to control group. This suggests a key role of PTH1R signaling in sustaining subchondral bone repairing by inhibiting excessive active TGF-β signaling47.

BMP pathway

The administration of iPTH to OVX mice, as first described by Ogtia in 2008, led to the rapid phosphorylation of SMAD1/5/8, which are downstream targets of BMP-2, in the periosteum. This, in turn, increase the osteogenic differentiation of SSCs72. Consistently, iPTH was also found to activate SMAD1 pathway in directly via inducing the endocytosis of LRP6. LRP6, a co-receptor in the Wnt signaling pathway, has been identified to interact with both the PTH and BMP extracellular signaling pathways. It forms a complex with the PTH1R and shares common antagonists with BMP signaling, suggesting a cross-talk or shared regulation between these pathways. Treatment with PTH induces the endocytosis of the PTH1R-LRP6 complex, and leads to an enhancement in the phosphorylation of SMAD1, resulting in enhanced differentiation of Sca-1(+) CD45(-) CD11b(-) SSCs into the osteoblast lineage. Importantly, this effect is abolished when PTH1R, or β-arrestin, is deleted or inhibited, indicating the involvement of these components in mediating the phosphorylation of SMAD1 in response to PTH treatment74.”

β2-AR section

The PTH1R-cAMP-PKA pathway, central to the cellular actions of PTH, has been found to be modulated by other receptors, including the β2 Adrenergic Receptor (β2-AR). This interaction suggests a cross-talk between different receptor systems, extending the complexity of how PTH influences bone metabolism. The research of Hanyu in 2012 indicates that iPTH lacks osteoanabolic activity in mice that do not have the β2-AR. It was discovered that the deficiency of β2-AR prevents the expression of iPTH-targeted genes that are involved in bone formation and resorption and are usually regulated by the cAMP/PKA signaling pathway117. Intriguingly, PTH treatment has been shown to quickly suppress the expression levels of β2-AR mRNA in the study of Moriya118, which subsequently lead to an increase in the phosphorylation of the CREB. Actually, β2AR has been reported to exert direct negative effects on bone mass by activating bone resorption and suppressing bone formation. One of the potential factors involved in the lack of PTH anabolism seen in β2AR-KO mice could be due to  the down-regulation of genes encoding proteins necessary for the osteoblastic function from other cell type117. These kind of intriguing properties and link between PTH treatment and β2-AR may also be one of their ways of desensitization and serve as a break for PTH signaling. Therefore, the β2-AR may play a critical role in regulating the effects of PTH1R-cAMP-PAK pathway on bone metabolism, but the exact mechanism is still unclear.

  1. The lack of effect of PTH on SSC senescence (line 426) contrasts with what is mentioned previously (line 355).

Response: Thank you for your corrections, we apologize for the mistake and revised it to “Consistent with most studies that PTH had limited effect on SSCs proliferation, PTH has also been shown to promote the proliferation of SSCs without affecting apoptosis but reducing cell senescence.” In line 426.

  1. The role of PTH-induced SIRT1 as a putative mechanism to prevent MMP13 overexpression in osteoblasts is unclear (lines 681-4).

Response: Thank you for your points, we have revised it to “On the contrary, SIRT1 acts as a feedback inhibitor of Mmp13 transcription. PTH treatment increases the interaction of SIRT1 with c-Jun, leading to the deacetylation of c-Jun. Subsequently, SIRT1-cJun complex is recruited to the AP-1 site on the Mmp13 promoter, which results in the inhibition of PTH-stimulated Mmp13 expression166.”

  1. The dual role of PTH through the PKA-SIK2 signaling pathway in bone remodeling (lines 781-96), as well as the true role of the Notch pathway in osteocytes as a mediator of PTH anabolism (lines 816-26), should be more elaborated.

Response: Thank you for your suggestions, we revised the related contents as followed:

PKA-SIK pathway

Salt-Inducible Kinases (SIKs) have been identified as crucial effectors within the PTH-PKA signaling pathway in osteocytes. In the absence of PTH signaling, SIK2 tonically phosphorylates its substrates HDAC4/5 and CRTC2, leading to their cytoplasmic retention via binding to 14-3-3 chaperones.  Upon activation by PTH, intracellular signaling leads to the activation of PKA, which subsequently phosphorylates and inactivates SIK2. When SIK2 is active and unphosphorylated, it phosphorylates histone deacetylases HDAC4 and HDAC5. However, when PTH-mediated signaling leads to the phosphorylation and inactivation of SIK2, which allows HDAC4 and HDAC5 to translocate into the nucleus. Within the nucleus, HDAC4 and HDAC5 act to inhibit the function of myocyte-specific enhancer factor 2C (MEF2C)189, a transcription factor that normally drives the expression of the Sost gene, which is known to encode a protein that inhibits bone formation. Simultaneously, the inactivated SIK2 reduces the phosphorylation of CREB-regulated transcription coactivator 2 (CRTC2) and leading to their dephosphorylation by an unknown phosphatase, therefore promotes its nuclear translocation. . Once in the nucleus, CRTC2 enhances the activity of CREB, a transcription factor that, among other things, increases the expression of the Rankl gene190-191. Thus, the PTH-PKA-SIK2 signaling axis has a dual role in bone metabolism: it indirectly promotes bone formation by reducing Sost expression and simultaneously enhances bone resorption through increased Rankl expression. SIK2 inhibition represents an intracellular mechanism to ensure that PTH signaling stimulates both bone formation and bone resorption.

Since SIK2 suppression leads to regulation of key PTH target genes in osteocytes, not surprisingly, small molecular SIK2 inhibitors such as YKL05099 mimics many of the effects of PTH in osteocytes and following in vivo administration189. Overall, these observations indicate that SIK inhibition is a key signaling mechanism used by PTH to accomplish its physiologic effects in osteocytes.”

Notch pathway

  • Recent studies have investigated the relationship between PTH treatment and the Notch signaling pathway in osteocytes, yielding contrasting findings. Early studies reported that PTH treatment stimulates osteoblastic Jagged1 mainly through the AC/PKA signaling pathway downstream of the PTH1R and subsequently promote osteogenesis in osteoblastic cells. However, newer studies demonstrated that a brief period (6 hours) of PTH administration downregulates the expression of Notch pathway genes, hinders the DNA binding of the key Notch transcription factor Rbpjκ, and diminishes Notch signaling like Notch2, an inhibitor on osteogenic function, in osteocytes, which seems to be crucial for PTH's anabolic effects195-196197. A possible explanation is that Jag1 induction is not sufficient to overcome the inhibitory effect of PTH on Rbpjκ-mediated signaling that occurs primarily in the osteoblastic lineage but might not take place in the bone marrow niche.

In contrast, other studies reported that prolonged iPTH treatment boosts the expression of Notch pathway components like Notch1 in osteocytes, contributing to a reduced bone resorption and increased bone formation198. The activation of Notch1 promote the expression of OPG and also inhibits SOST in osteocytes, which increases bone formation199. The length of the treatment (30 days vs. hours) and the model used (mice vs. bone organ cultures) may account for the differences between these two groups of studies.”

  1. The studies involving SOST should be included in the Wnt pathway section (line 797). PTH inhibits not induces SOST (line 853).

Response: We agree with your suggestions and moved the SOST section to be part of the Wnt pathway section as followed:

Wnt pathway

Besides, Sclerostin is a well-established inhibitor of the Wnt signaling pathway, which is vital for bone formation. Normally, the administration of PTH leads to a reduction in sclerostin levels in osteocytes, thereby facilitating Wnt signaling and promoting bone formation202. This is highlighted by the observation that in animals with Ocy-PTH1RcKO, PTH fails to suppress the expression of Sost and its protein product sclerostin200. And it was suggested that the downregulation of Sost, leading to the activation of the Wnt pathway, is necessary for the stimulating impact of PTH1R signaling in osteocytes and also in DMP1-caPTHR1 mice201. Mechanistically, PTH induces the nuclear accumulation of histone deacetylases HDAC4 and HDAC5, which result from the inhibition of SIK2. The nuclear translocation of HDAC4/5 leads to the downregulation of myocyte enhancer factor 2 (MEF2), a transcription factor that typically upregulates Sost expression127, 203 204. In contrast, the suppression of Sost and MEF2 by PTH, along with the changes in HDAC levels, is markedly less effective in mice that lack the LRP6 receptor203 204-205. This finding indicates the essential role of LRP6 in facilitating the PTH-driven signaling that leads to the suppression of sclerostin, highlighting LRP6's importance in PTH's bone anabolic effects. Moreover, CDH2 has been identified as a protein that negatively influences the interaction between LRP6 and the PTH1R receptor, which is a necessary step for PTH to exert its effects206. By modulating the LRP6-PTHR1 interaction, N-cadherin indirectly mediates the suppressive action of PTH on Sost expression.  

In addition to SIK2-HDAC4/5 and LRP6-Wnt regulation, Periostin (Postn)207-208 and Tgif1209 were also considered to be the PTH1R-sinagling targets and involved in the PTH inhibited expression of Sost. Recombinant periostin has been found to suppress Sost expression, a process that is mediated through the integrin αVβ3 receptor. Conversely, using a periostin-blocking antibody has been shown to prevent the inhibition of MEF2C and Sost by PTH208. Moreover, upon PTH treatment, Tgif1 is upregulated via the PKA–pCREB and prompted to disengage from the SOST promoter, which also leads to a downregulation of Mef2c expression, resulting in decreased sclerostin production209.”

  1. T cells subsection (line 859) is misplaced under the Osteocytes section (line 772).

Response: Thank you for your valuable suggestions. We added “(3) Osteocytes-T cells cross talk” part in section 3.2 to discuss about osteocytes-t cell cross talk.

  1. RANKL acts on preosteocalsts not osteoblasts to secrete SAA3 in Choudhary et al. (ref. 232, line 956).

Response: Thank you for your corrections. We apologize for the mistake and corrected to preosteoclasts.

  1. The PTH actions described in sections 3.3 (Adipocytes) and 3.4 (Endothelial cells) are unrelated to PTH catabolism in bone.

Response: Thank you for your corrections. The section “3.2.2 Cellular actions of PTH contributing to increased bone resorption: Catabolic” was part of the section “3.2 Osteoblasts and Osteocytes ”, and sections 3.3 (Adipocytes) and 3.4 (Endothelial cells) are in parallel with “3.2 Osteoblasts and Osteocytes ”, not related to PTH catabolism in bone section.

  1. PTH has been shown to stimulate MCP-1 in various cell types (kidney, osteoblasts,...); thus, results described in lines 1068-71 are not really unexpected.

Response: Thank you for your corrections. We change the related description to “Moreover, PTHrP has been found to stimulate the expression of both the protein and mRNA of MCP-1 in human BM ECs through mechanisms dependent on the binding elements of C/EBPβ) and NF-κB. This process, in turn, promotes the migration and infiltration of monocytes and enhance the differentiation of macrophages253

  1. It is now known that interaction of PTH and PTHrP with different PTH1R conformations might help explain differences in signaling pathways leading to a predominant biological response (anabolic vs catabolic). This particular aspect should be included in this review.

Response: Thank you for your suggestions, we added related contents in the section 3.2 osteoblasts and osteocytes as followed:

3.2 Osteoblasts and Osteocytes

The levels of PTH in the bloodstream can lead to either catabolic (bone-resorbing) or anabolic (bone-forming) effects on bone, depending on the timing and pattern of the hormone's elevation. The anabolic effects of iPTH treatment are attributed to several concurrent mechanisms. These include the activation of immediate-early genes, a rise in the expression and/or function of key osteoblast transcription factors, and the suppression of proteins that negatively impact osteogenesis, like sclerostin. In contrast, continuous PTH administration primarily causes bone loss by increasing the expression of RANKL, a promoter of osteoclast activity, and decreasing the expression of osteoprotegerin, which is a natural inhibitor of bone resorption86.

Moreover, it is well-known that PTH and PTHrP are key regulators in calcium and phosphate metabolism, as well as bone remodeling. Similar to SSCs, upon the binding of PTH or PTHrP to the PTH1R receptor, a cascade of specific responses is initiated at the level of G Proteins, and subsequently trigger different biological responses. While the activation of most PTH-responsive genes is a result of the combined action of Gs, Gq and G12/13 proteins, a distinct set of genes is regulated by the trio of Gs, Gq and G12/13, including Gsα-cAMP-PKA pathway, Gq-PLC-β-PKC pathway and Gα12/13-phospholipase D-transforming protein RhoA pathway101-102. It has been reported that varying effective domains within PTH or PTHrP are responsible for the differential activation of the PTH1R receptor, which in turn selectively triggers downstream signaling pathways103-104,105-106. Overall, this interaction of PTH and PTHrP with different PTH1R conformations may help explain the differences in signaling pathways leading to a predominant anabolic and catabolic biological response.”

Other points.-

- Acronyms must be described the first time mentioned.

Response: Thank you for your corrections. We revised as requested.

- The term "non-redundant" (line 147) is not accurate, since PTHrP is the PTH counterpart in bone tissue (not "a hormone", line 210).

Response: Thank you for your corrections. We revised related description to “PTH is an essential and key modulator of…”.

- The first author of an article must be followed by "et al" when referenced in the text. Figures citation should be included where corresponding.

Response: Thank you for your corrections. We revised as requested.

Reviewer 3 Report

Comments and Suggestions for Authors

The authors emphasize the crucial involvement of parathyroid hormone (PTH) in maintaining calcium homeostasis, primarily through its modulation of bone remodeling processes. The impact of PTH on bone is intricately tied to the duration and frequency of exposure. Depending on whether the administration is continuous or intermittent, PTH can instigate both bone formation and resorption. Continuous PTH administration appears to lean towards promoting bone resorption, potentially by regulating specific genes within bone cells. Conversely, intermittent exposure generally supports bone formation, potentially through transient gene activation.

PTH's influence extends across various facets of bone cell activity, directly affecting skeletal stem cells, cells in the osteoblastic lineage, osteocytes, and T cells, thereby playing a critical role in bone generation. Concurrently, it indirectly influences osteoclast precursor cells and osteoclasts, while also directly impacting T cells, contributing to its involvement in bone resorption. Despite these insights, the intricate mechanisms through which PTH operates within the bone marrow niche remain not entirely understood.

The article comprehensively reviews the dual roles of PTH—catabolic and anabolic—on bone cells, highlighting the intricate cellular and molecular pathways involved in these processes. The complex interplay of these factors in bone remodelling underscores the necessity for further investigation to fully grasp PTH's multifaceted influence on bone health.

The initial perception of PTH's role in the skeleton primarily positioned it as a promoter of bone resorption. However, it is now acknowledged that PTH can indeed stimulate bone formation. The ultimate impact of PTH on bone mass, whether it is anabolic or catabolic, depends on the balance between these processes. The determination of PTH's effect is contingent on which of these actions, bone formation or resorption, is predominantly influenced by its activity on mesenchymal cells within the BM niche. In summary, continuous exposure to high levels of PTH is typically associated with bone degradation, while periodic exposure to low doses has been found to promote bone growth.

In intermittent administration, PTH triggers the transformation of SSCs into osteoblasts, enhances the growth and maturation of osteoblasts or osteocytes, and significantly fosters bone formation. Mechanistically, iPTH binds to PTH1R, activating Gαs in SSCs, osteoblasts, and osteocytes, leading to the production of cAMP and initiation of the PKA cascade reaction. PKA, in turn, promotes the phosphorylation of downstream targets, inducing osteogenic gene transcription and inhibiting gene expression of Sost, thereby increasing osteogenesis and cell migration. Additionally, iPTH enhances the levels of IGF1, Amphiregulin, Wnt10b, and BMPs, activating their respective receptors and initiating downstream signaling pathways.

With continuous administration, PTH indirectly influences osteoclast formation through its impact on osteoblasts and osteocytes, encouraging their growth, development, and cell fusion, ultimately leading to increased bone resorption. The mechanism involves PTH1R triggering intracellular events, phosphorylating CREB, inhibiting HDAC2, and enhancing the transcriptional activity of MEF2C, resulting in increased gene expression of Rankl and Mab21. Moreover, cPTH stimulates osteoblasts and osteocytes to produce factors that play a crucial role in promoting the differentiation and cell fusion of osteoclast precursors.

The regulation of bone by PTH involves not only cells of osteoblastic lineage but also T cells from the immune system, highlighting the complexity of PTH's role in bone regulation. There is a reciprocal regulatory relationship between immune and bone cells, indicating a significant interplay between these two systems in the context of PTH's function. T cells, in response to PTH treatment, can produce either Wnt10b or Th17α, influencing both the anabolic and catabolic actions of PTH within the BM. B cells have also been found to produce RANKL, suggesting their involvement in bone remodeling mediated by PTH.

Further suggestions for improvement:

  1. Clarity and Conciseness:

    • Consider breaking down complex sentences into simpler ones to enhance readability.
    • Aim for clarity in conveying the main points, avoiding overly complex language.
  2. Structural Organization:

    • Add section headings to organize the discussion into key themes, making it easier for readers to follow the flow of information.
    • Use bullet points or numbered lists for sequences of events or processes to enhance clarity.
  3. Transition Phrases:

    • Use transition phrases to smoothly guide readers from one idea to the next, improving the overall coherence of the text.
  4. Avoid Redundancy:

    • Identify and eliminate any redundant information to maintain a streamlined narrative.
  5. Citations:

    • Ensure that all relevant information is properly cited to give credit to the original sources and enhance the credibility of the discussion.
  6. Definition of Terms:

    • Include brief explanations or definitions for specialized terms to make the content more accessible to a wider audience.
  7. Visual Aids:

    • Consider incorporating figures, diagrams, or tables to visually represent complex processes, adding another layer of understanding for readers.
  8. Consistent Terminology:

    • Ensure consistent use of terminology throughout the discussion to avoid confusion.
  9. Active Voice:

    • Where possible, use the active voice to create a more engaging and direct writing style"i
    • n conclusion, this review compiles a wide array of research highlighting the significance of PTH1R signaling in the regulation of mesenchymal cells within the BM niche. Despite the clear therapeutic benefits of PTH-related treatments, ongoing research is needed to fully elucidate the mechanisms of action and optimize therapeutic strategies for individual patients. The critical function of PTH1R in bone remodeling and bone physiology is underscored, along with the therapeutic potentials of PTH and PTHrP in the management of osteoporosis.
  10. Finally, an
  11. extended discussion could include the impact of skeletal metastasis of unknown primary origin on parathyroid hormone (PTH) dynamics and bone health. Emphasizing the importance of understanding PTH's interaction with bone cells within the tumor microenvironment, the discourse highlights the complexities associated with managing complications arising from metastatic lesions. The clinical implications of PTH-related therapies in the context of skeletal metastasis underscore the need for personalized and targeted approaches. The section also calls for further research to unravel specific mechanisms and guide future advancements in clinical care. Overall, the inclusion of skeletal metastasis of unknown primary enriches the discussion, providing a comprehensive perspective on the interplay between PTH, bone health, and metastatic diseases. (

SMUP; bisphosphonates; bone markers; bone metastases; skeletal metastases of unknown primary; tumor microenvironment; unknown primary tumor are crucial topic and available review and original manuscript can be considered in the discussion sections.)

Comments on the Quality of English Language

The authors emphasize the crucial involvement of parathyroid hormone (PTH) in maintaining calcium homeostasis, primarily through its modulation of bone remodeling processes. The impact of PTH on bone is intricately tied to the duration and frequency of exposure. Depending on whether the administration is continuous or intermittent, PTH can instigate both bone formation and resorption. Continuous PTH administration appears to lean towards promoting bone resorption, potentially by regulating specific genes within bone cells. Conversely, intermittent exposure generally supports bone formation, potentially through transient gene activation.

PTH's influence extends across various facets of bone cell activity, directly affecting skeletal stem cells, cells in the osteoblastic lineage, osteocytes, and T cells, thereby playing a critical role in bone generation. Concurrently, it indirectly influences osteoclast precursor cells and osteoclasts, while also directly impacting T cells, contributing to its involvement in bone resorption. Despite these insights, the intricate mechanisms through which PTH operates within the bone marrow niche remain not entirely understood.

The article comprehensively reviews the dual roles of PTH—catabolic and anabolic—on bone cells, highlighting the intricate cellular and molecular pathways involved in these processes. The complex interplay of these factors in bone remodelling underscores the necessity for further investigation to fully grasp PTH's multifaceted influence on bone health.

The initial perception of PTH's role in the skeleton primarily positioned it as a promoter of bone resorption. However, it is now acknowledged that PTH can indeed stimulate bone formation. The ultimate impact of PTH on bone mass, whether it is anabolic or catabolic, depends on the balance between these processes. The determination of PTH's effect is contingent on which of these actions, bone formation or resorption, is predominantly influenced by its activity on mesenchymal cells within the BM niche. In summary, continuous exposure to high levels of PTH is typically associated with bone degradation, while periodic exposure to low doses has been found to promote bone growth.

In intermittent administration, PTH triggers the transformation of SSCs into osteoblasts, enhances the growth and maturation of osteoblasts or osteocytes, and significantly fosters bone formation. Mechanistically, iPTH binds to PTH1R, activating Gαs in SSCs, osteoblasts, and osteocytes, leading to the production of cAMP and initiation of the PKA cascade reaction. PKA, in turn, promotes the phosphorylation of downstream targets, inducing osteogenic gene transcription and inhibiting gene expression of Sost, thereby increasing osteogenesis and cell migration. Additionally, iPTH enhances the levels of IGF1, Amphiregulin, Wnt10b, and BMPs, activating their respective receptors and initiating downstream signaling pathways.

With continuous administration, PTH indirectly influences osteoclast formation through its impact on osteoblasts and osteocytes, encouraging their growth, development, and cell fusion, ultimately leading to increased bone resorption. The mechanism involves PTH1R triggering intracellular events, phosphorylating CREB, inhibiting HDAC2, and enhancing the transcriptional activity of MEF2C, resulting in increased gene expression of Rankl and Mab21. Moreover, cPTH stimulates osteoblasts and osteocytes to produce factors that play a crucial role in promoting the differentiation and cell fusion of osteoclast precursors.

The regulation of bone by PTH involves not only cells of osteoblastic lineage but also T cells from the immune system, highlighting the complexity of PTH's role in bone regulation. There is a reciprocal regulatory relationship between immune and bone cells, indicating a significant interplay between these two systems in the context of PTH's function. T cells, in response to PTH treatment, can produce either Wnt10b or Th17α, influencing both the anabolic and catabolic actions of PTH within the BM. B cells have also been found to produce RANKL, suggesting their involvement in bone remodeling mediated by PTH.

Further suggestions for improvement:

  1. Clarity and Conciseness:

    • Consider breaking down complex sentences into simpler ones to enhance readability.
    • Aim for clarity in conveying the main points, avoiding overly complex language.
  2. Structural Organization:

    • Add section headings to organize the discussion into key themes, making it easier for readers to follow the flow of information.
    • Use bullet points or numbered lists for sequences of events or processes to enhance clarity.
  3. Transition Phrases:

    • Use transition phrases to smoothly guide readers from one idea to the next, improving the overall coherence of the text.
  4. Avoid Redundancy:

    • Identify and eliminate any redundant information to maintain a streamlined narrative.
  5. Citations:

    • Ensure that all relevant information is properly cited to give credit to the original sources and enhance the credibility of the discussion.
  6. Definition of Terms:

    • Include brief explanations or definitions for specialized terms to make the content more accessible to a wider audience.
  7. Visual Aids:

    • Consider incorporating figures, diagrams, or tables to visually represent complex processes, adding another layer of understanding for readers.
  8. Consistent Terminology:

    • Ensure consistent use of terminology throughout the discussion to avoid confusion.
  9. Active Voice:

    • Where possible, use the active voice to create a more engaging and direct writing style"i
    • n conclusion, this review compiles a wide array of research highlighting the significance of PTH1R signaling in the regulation of mesenchymal cells within the BM niche. Despite the clear therapeutic benefits of PTH-related treatments, ongoing research is needed to fully elucidate the mechanisms of action and optimize therapeutic strategies for individual patients. The critical function of PTH1R in bone remodeling and bone physiology is underscored, along with the therapeutic potentials of PTH and PTHrP in the management of osteoporosis.
  10. Finally, an
  11. extended discussion could include the impact of skeletal metastasis of unknown primary origin on parathyroid hormone (PTH) dynamics and bone health. Emphasizing the importance of understanding PTH's interaction with bone cells within the tumor microenvironment, the discourse highlights the complexities associated with managing complications arising from metastatic lesions. The clinical implications of PTH-related therapies in the context of skeletal metastasis underscore the need for personalized and targeted approaches. The section also calls for further research to unravel specific mechanisms and guide future advancements in clinical care. Overall, the inclusion of skeletal metastasis of unknown primary enriches the discussion, providing a comprehensive perspective on the interplay between PTH, bone health, and metastatic diseases. (

SMUP; bisphosphonates; bone markers; bone metastases; skeletal metastases of unknown primary; tumor microenvironment; unknown primary tumor are crucial topic and available review and original manuscript can be considered in the discussion sections.)

Author Response

Further suggestions for improvement:

Clarity and Conciseness:

Consider breaking down complex sentences into simpler ones to enhance readability.

Aim for clarity in conveying the main points, avoiding overly complex language.

Structural Organization:

Add section headings to organize the discussion into key themes, making it easier for readers to follow the flow of information.

Use bullet points or numbered lists for sequences of events or processes to enhance clarity.

Transition Phrases:

Use transition phrases to smoothly guide readers from one idea to the next, improving the overall coherence of the text.

Avoid Redundancy:

Identify and eliminate any redundant information to maintain a streamlined narrative.

Citations:

Ensure that all relevant information is properly cited to give credit to the original sources and enhance the credibility of the discussion.

Definition of Terms:

Include brief explanations or definitions for specialized terms to make the content more accessible to a wider audience.

Visual Aids:

Consider incorporating figures, diagrams, or tables to visually represent complex processes, adding another layer of understanding for readers.

Consistent Terminology:

Ensure consistent use of terminology throughout the discussion to avoid confusion.

Active Voice:

Where possible, use the active voice to create a more engaging and direct writing style"

Response: Thank you for your suggestions, we revised our manuscript as requested to make our manuscript more readable.

Finally, an extended discussion could include the impact of skeletal metastasis of unknown primary origin on parathyroid hormone (PTH) dynamics and bone health. Emphasizing the importance of understanding PTH's interaction with bone cells within the tumor microenvironment, the discourse highlights the complexities associated with managing complications arising from metastatic lesions. The clinical implications of PTH-related therapies in the context of skeletal metastasis underscore the need for personalized and targeted approaches. The section also calls for further research to unravel specific mechanisms and guide future advancements in clinical care. Overall, the inclusion of skeletal metastasis of unknown primary enriches the discussion, providing a comprehensive perspective on the interplay between PTH, bone health, and metastatic diseases. (SMUP; bisphosphonates; bone markers; bone metastases; skeletal metastases of unknown primary; tumor microenvironment; unknown primary tumor are crucial topic and available review and original manuscript can be considered in the discussion sections.)

Response:Thank you for your valuable suggestions, we added related contents in our conclusion part as followed:

“In addition to normal cells, it was reported that tumor cells could secrete PTHrP, which stimulates bone resorption via increasing RANKL expression in osteoblastic cells255. Besides, the produced PTHrP by tumor cells could also acts on osteoblastic cells to increase expression of Jagged 1, which may promote angiogenesis by activating Notch signaling on HSCs and support the tumor cell niche in bone256. This interaction distinctively enhances signals beneficial for tumor localization in bone. While PTHrP is recognized as a harmful factor in tumor development within metastatic lesions, more research is required to accurately determine the relationship between PTHrP and tumor progression in humans255, 257

Reviewer 4 Report

Comments and Suggestions for Authors

Dear authors,

the review is comprehensive, thank you. You mention vast number of genes and factors, and, as it is a review, I would ask to add the explanations for the abbreviations and a few words for the functions of the gene/protein in question. You've done it in some fragments of the text, but not everywhere.

Why do you use the term 'skeletal stem cells' instead of much more common 'mesenchymal stem cells'? 

Comments on the Quality of English Language

Following lines have some typos or grammar mistakes, please, correct: 331, 451, 500, 516, 611, 738, 826, 844, 931, 942, 945.

Author Response

  1. the review is comprehensive, thank you. You mention vast number of genes and factors, and, as it is a review, I would ask to add the explanations for the abbreviations and a few words for the functions of the gene/protein in question. You've done it in some fragments of the text, but not everywhere.

Response: Thank you for your corrections. We revised as requested.

  1. Why do you use the term 'skeletal stem cells' instead of much more common 'mesenchymal stem cells'? 

Response: SSCs, or skeletal stem cells are now recognized as the most appropriate term for MSCs since these cells are self-renewing and have trilinear potential to become adipocytes, chondrocytes or osteoblasts. Please see Nature Reviews Endocrinology Horowitz MC and Rosen CJ. 2023 Nov;19(11):626-638. doi: 10.1038/s41574-023-00879-4. Epub 2023 Aug 16.

  1. Following lines have some typos or grammar mistakes, please, correct: 331, 451, 500, 516, 611, 738, 826, 844, 931, 942, 945.

Response: Thank you for your corrections. We revised as requested.

Round 2

Reviewer 2 Report

Comments and Suggestions for Authors

Most of the raised concerns have been appropriately addressed in the revised version. However, some specific points described below still deserve consideration in order to improve the quality of the manuscript:

- Various references in paragraph 3.1.1. correspond to in vivo studies in humans or animals; thus, I do not see the reason to include a separate subsection dealing with this type of studies.

- The putative impact of PTH-induced hematopoiesis from SSCs on osteogenic differentiation is not obvious and deserves further clarification (paragraph 3.1.6).

- The sections 3.2.1 and 3.2.2 contain various subsections without numbering. The latter facilitates reading the different sections dealing with many related aspects.

- Osteocytes-T cells section does not seem to include osteocyte-specific mechanisms. This paragraph might also be part of the Osteoblast signaling section.

Other points:

- PTHrP is not a hormone (paragraph 3.1.2. Adipogenesis) but rather a local factor.

- Reference citation must occur sequentially in order of appearance in the text.

Comments on the Quality of English Language

English language still requires careful revision.

Author Response

Most of the raised concerns have been appropriately addressed in the revised version. However, some specific points described below still deserve consideration in order to improve the quality of the manuscript:

- Various references in paragraph 3.1.1. correspond to in vivo studies in humans or animals; thus, I do not see the reason to include a separate subsection dealing with this type of studies.

Response: thank you for your advice, we decided to remove that part about in vivo studies.

- The putative impact of PTH-induced hematopoiesis from SSCs on osteogenic differentiation is not obvious and deserves further clarification (paragraph 3.1.6).

Response: thank you for your suggestions, we added a more detailed discussion about the effect of PTH-induced hematopoiesis on SSCs as follows:

“3.1.6 Indirect effect of PTH on hematopoiesis via regulating SSCs

PTH also exerts an indirect effect on hematopoiesis through its actions on SSCs. PTH may boost hematopoiesis by activating the IGF system, such as increasing the secretion of Insulin-like growth factor (IGF)-2, Insulin-like growth factor binding protein (IGFBP)-1, 2, 3 and GM-CSF in SSCs57. PTH stimulates the expansion of CD34(+) HSCs and supports hematopoiesis by upregulating the expression level of CDH11 in BM-derived SSCs58. Therefore, PTH treatment may serve as an effective strategy to enhance the ability of BM-derived SSCs to support hematopoiesis. Conversely, PTH can also impact CD8+ T cells by increasing their expression of key Wnt factors, including Wnt5a, Wnt10b, Wnt6, and Wnt10a59. PTH has been reported to increase the levels of the soluble interleukin-6 receptor (sIL-6R) in primary osteoblast cultures. While sIL-6R does not directly influence osteoblast proliferation or the differentiation of SSCs in vitro, it plays a significant role in the BM microenvironment by enhancing the expansion of myeloid cells and recruiting SSCs via increasing the production of TGF-β1in SSCs60.

During the development of both mice and humans, MSCs and hematopoietic cells coexist in most hematopoietic tissues, from the embryonic stage to adulthood61. This coexistence suggests a close relationship between these two cell types. Numerous studies have shown that the state of HSCs is intimately linked to bone health62. Specifically, bone marrow erythroid progenitor cells can influence the differentiation of MSCs into osteoblasts and adipocytes through direct cell-to-cell contact63. Rodrigo et al. found that reduced levels of hemoglobin and blood cells might be early indicators of bone fragility in men64. In laboratory experiments, it was observed that a low concentration of HSCs in co-culture with MSCs can accelerate the osteogenic progression of MSCs. This acceleration was evidenced by an earlier peak in alkaline phosphatase activity and increased calcium deposition, compared to cultures of MSCs alone65. Further studies confirmed that HSCs can regulate the induction of mesenchymal stromal cells into osteoblasts. This regulation is part of the formation of the stem cell niche and is facilitated by the secretion of Bone Morphogenetic Proteins 2 and 6 (BMP2 and BMP6)66. These findings highlight the complex interplay between different cell types in bone health and development. Overall, the indirect effect on hematopoiesis has a beneficial effect on SSCs osteogenic differentiation, thereby supporting the differentiation of osteoprogenitors and contributing to the anabolic bone remodeling effects of PTH59, 67, 68.”

- The sections 3.2.1 and 3.2.2 contain various subsections without numbering. The latter facilitates reading the different sections dealing with many related aspects.

Response: thank you for your advice, we changed the related section number to make it more readable.

- Osteocytes-T cells section does not seem to include osteocyte-specific mechanisms. This paragraph might also be part of the Osteoblast signaling section.

Response:thank you for your suggestion, we changed it to “3.2.3 The downstream regulatory networks of PTH that are anabolic in T cells

Other points:

- PTHrP is not a hormone (paragraph 3.1.2. Adipogenesis) but rather a local factor.

Response:thank you for your correction, we changed it to “Parathyroid hormone-related protein (PTHrP) (1-36), another local factor that shares structural similarities with PTH…”.

- Reference citation must occur sequentially in order of appearance in the text.

Response:thank you for your suggestion, we checked again to make the reference citation occur in order.

Reviewer 3 Report

Comments and Suggestions for Authors

The authors responded to the revision in a short rebuttal. It would be great to provide a point-by-point rebuttal, a marked-up version of the revised manuscript and a complete selection of the expanded revised references including the citing manuscripts on the topic expanded in the discussion *i.e. all the manuscripts dealing with SMUP, including review, mata-analyses and original manuscripts.

Comments on the Quality of English Language

The authors responded to the revision in a short rebuttal. It would be great to provide a point-by-point rebuttal, a marked-up version of the revised manuscript and a complete selection of the expanded revised references including the citing manuscripts on the topic expanded in the discussion *i.e. all the manuscripts dealing with SMUP, including review, mata-analyses and original manuscripts.

Author Response

The authors responded to the revision in a short rebuttal. It would be great to provide a point-by-point rebuttal, a marked-up version of the revised manuscript and a complete selection of the expanded revised references including the citing manuscripts on the topic expanded in the discussion *i.e. all the manuscripts dealing with SMUP, including review, mata-analyses and original manuscripts.

Response:thank you for your suggestion, we expanded the discussion about SMUP and PTHrP as follows:

  • PTHrP as the therapeutic target for Skeletal Metastasis of Unknown Primary (SMUP)

Skeletal metastases of unknown primary (SMUP) represent enigmatic rare metastatic tumor entities without anatomic primary sites identified. Mechanically, the process of cancer spreading to the bone is the homing to the marrow microenvironment throughout the bloodstream of the tumor cells, via the neo-angiogenic process, and through permissive bone marrow endothelial cells260, 261. Recently, a number of molecular actors have been considered as elements that drive neoplastic cells to the bone environment, including adhesion molecule systems such as CXCL12/CXCR4 (chemokine ligand 12/chemokine receptor 4)262, junctional adhesion molecules in osteotropic tumors 263, 264, focal adhesion kinases265, 266, and vascular and immune-microenvironment interactions267, 268. Once cancer cells reach the bone marrow , they cross the wall of sinusoids, invade the matrix, and, once they reach the endosteal surface, stimulate osteoclastogenic activity and proliferate with the formation of metastases with prevailing osteolytic or osteoblastic development. These extremes cover the spectrum of neoplastic bone remodeling269.

Numerous clinical trials are currently exploring various molecules that impact the progression of bone metastases in solid tumors, either directly or indirectly. Some of these molecules specifically target the bone resorption process by focusing on certain bone cells like osteoclasts, osteoblasts, osteocytes, or by influencing molecular pathways that regulate these cells' functions270. Key molecules under investigation include denosumab271, inhibitors of endothelin 1 receptor, drugs that disrupt the Wnt/dkk1 pathway272, and src inhibitors273. However, the primary focus of these drugs so far has been their ability to halt the destruction of the bone matrix caused by tumor cells. These treatments aim to intervene in the complex interactions between cancer progression and bone health, potentially offering new avenues for managing bone metastases in cancer patients. Furthermore, emerging research is lending support to the theory that PTHrP plays a significant role in influencing the bone microenvironment. It has emerged as a key player in various cancers, particularly in their skeletal metastasis274-276. It affects cells within the bone microenvironment, exerting proliferative and pro-survival effects that prepare both the 'seed' (the cancer cells) and the 'soil' (the bone environment) for the establishment and growth of metastatic lesions277. This new understanding of PTHrP's role highlights the complex dynamics between cancer progression and bone biology, opening up potential avenues for targeted therapies in oncology.